# Immune Modulation in Alzheimer’s Disease: From Pathogenesis to Immunotherapy

**DOI:** 10.3390/cells14040264

**Published:** 2025-02-12

**Authors:** Sahar Balkhi, Anna Di Spirito, Alessandro Poggi, Lorenzo Mortara

**Affiliations:** 1Immunology and General Pathology Laboratory, Department of Biotechnology and Life Sciences, University of Insubria, 21100 Varese, Italy; sahar.balkhi@uninsubria.it (S.B.); adispirito@uninsubria.it (A.D.S.); lorenzo.mortara@uninsubria.it (L.M.); 2Molecular Oncology and Angiogenesis Unit, IRCCS Ospedale Policlinico San Martino, 16132 Genoa, Italy

**Keywords:** Alzheimer’s disease, microglial activation, chronic neuroinflammation, immunotherapy

## Abstract

Alzheimer’s disease (AD) is a progressive neurodegenerative disorder and the leading cause of dementia, affecting a significant proportion of the elderly population. AD is characterized by cognitive decline and functional impairments due to pathological hallmarks like amyloid β-peptide (Aβ) plaques and neurofibrillary tangles (NFTs) composed of hyperphosphorylated tau. Microglial activation, chronic neuroinflammation, and disruptions in neuronal communication further exacerbate the disease. Emerging research suggests that immune modulation could play a key role in AD treatment given the significant involvement of neuroinflammatory processes. This review focuses on recent advancements in immunotherapy strategies aimed at modulating immune responses in AD, with a specific emphasis on microglial behavior, amyloid clearance, and tau pathology. By exploring these immunotherapeutic approaches, we aim to provide insights into their potential to alter disease progression and improve patient outcomes, contributing to the evolving landscape of AD treatment.

## 1. Introduction

Alzheimer’s disease (AD) stands as a progressive and irreversible neurodegenerative disorder primarily associated with aging, serving as the leading cause of dementia worldwide. It imposes significant cognitive and functional impairments on individuals, particularly in the elderly. Approximately 13% of people aged 75 to 84 years and 33.3% of those aged 85 or older are affected by AD, making it a pressing public health concern. In the United States, AD ranks as the fifth leading cause of death among individuals aged 65 and older, with projections estimating that by 2050, around 131.5 million people globally will be living with the condition [1,2,3].

The classification of AD includes early-onset Alzheimer’s disease (EOAD), late-onset Alzheimer’s disease (LOAD), and sporadic Alzheimer’s disease. These categories are defined based on the age at which the disease manifests and the associated genetic factors. EOAD is diagnosed before the age of 65 and accounts for less than 5% of all Alzheimer’s cases. It is often associated with genetic mutations in the APP, PSEN1, and PSEN2 genes, which lead to the accumulation of amyloid β plaques and result in rapid cognitive decline. Symptoms commonly include aphasia and executive dysfunction. In contrast, LOAD is the most prevalent form of AD, typically developing after the age of 65. LOAD is characterized by a slower progression, with memory deficits being the primary symptom. It is strongly linked to the APOE ε4 allele as a significant genetic risk factor. Sporadic AD, which is generally considered synonymous with LOAD, arises from a combination of aging, lifestyle factors, and comorbidities such as cardiovascular disease. Unlike EOAD, it does not exhibit a clear familial inheritance pattern but shares similar pathological features, including amyloid β plaques, tau tangles, and neuroinflammation. Together, these variations highlight the complex interplay among genetic factors, environmental influences, and neurodegenerative processes central to the pathophysiology of AD [4,5]

The clinical manifestations of AD typically begin with gradual cognitive decline, characterized by memory loss, language difficulties, impaired reasoning, and personality changes. These symptoms worsen over time, leading to severe cognitive and functional impairment. AD progression is commonly categorized into three stages: the preclinical phase, where brain changes occur without noticeable behavioral symptoms; mild cognitive impairment, marked by mild memory lapses and word-finding difficulties; and the dementia stage, where individuals may struggle to recognize loved ones and perform basic daily tasks [1,2,3].

Pathologically, AD is defined by the accumulation of extracellular amyloid β-peptide (Aβ) and intracellular neurofibrillary tangles (NFTs) composed of hyperphosphorylated tau protein. Aβ and tau act synergistically to drive the progression of AD, with Aβ often considered the “trigger” and tau the “bullet” in this process. Soluble Aβ species initiate tau pathology, while tau amplifies the neurotoxic effects of Aβ, leading to synaptic loss and cognitive decline. This interplay exacerbates neuronal damage, oxidative stress, synaptic dysfunction, and neuroinflammation, ultimately resulting in the loss of synapses and neurons [6,7,8,9].

Microglial activation plays a significant role in mediating these pathological effects. Oligomeric Aβ promotes microglial activation, which contributes to early synaptic loss. Moreover, microglia may facilitate the spread of tau pathology, rendering synapses more vulnerable to tau-induced impairments. This neuroinflammatory cascade leads to neuronal cell death, plasma leakage, and brain atrophy. These pathological changes disrupt communication between brain cells, impairing cognitive functions such as memory and reasoning [9,10].

A potential link between amyloid and tau pathology may also involve the innate immune system, with both proteins triggering immune responses that further propagate damage. Together, these processes create a cascade of neuronal dysfunction and degeneration that underpins the clinical symptoms of AD [11].

Additionally, while extracellular Aβ plaques are long-recognized pathological hallmarks, increasing evidence highlights that Aβ aggregates within neurons before extracellular plaque formation also contribute to early synaptic dysfunction and cognitive decline.

Aβ peptides, particularly Aβ42, accumulate intracellularly in organelles such as the endoplasmic reticulum, Golgi apparatus, and endosomal–lysosomal systems [12,13,14]. This intraneuronal Aβ is associated with disrupted neuronal function, including synaptic degradation, impaired axonal transport, and loss of microtubule-associated proteins like MAP2. Notably, Aβ oligomers, the soluble precursors of plaques, are particularly neurotoxic, causing structural alterations in synapses and neurites [15]. Immunoelectron microscopy studies have demonstrated that intraneuronal Aβ localizes within multivesicular bodies and other vesicles, suggesting a role in the dysregulation of cellular trafficking. Additionally, intracellular Aβ appears to influence tau pathology, potentially acting as a precursor to neurofibrillary tangles [16,17].

The interplay between intracellular and extracellular Aβ is significant. Intraneuronal Aβ may seed extracellular plaque formation through neuritic degeneration, while extracellular Aβ can be internalized, exacerbating intracellular accumulation. These findings challenge the traditional focus on extracellular plaques and suggest that targeting intraneuronal Aβ might provide novel therapeutic opportunities.

Moreover, microglia, the resident immune cells in the brain, play a complex role in AD. Traditionally categorized into pro-inflammatory M1 and neuroprotective M2 types, recent studies reveal a spectrum of microglial phenotypes in AD, including disease-associated microglia (DAM), proliferative-region-associated microglia (PAM), and microglial neurodegenerative phenotype (MGnD). Each subtype has distinct gene expressions and functions, influencing the progression of the disease [18,19,20,21,22,23,24].

In the early stages of AD, microglia exhibit neuroprotective functions, primarily through the clearance of Aβ. However, as the disease progresses, microglia transition from a homeostatic state to an acute inflammatory state, and eventually to disease-associated microglia (DAM) or microglia in neurodegenerative diseases (MGnD). In this process, microglia initially engage in Aβ clearance, but as activation persists, they adopt a pro-inflammatory phenotype that exacerbates neuroinflammation and neuronal damage. Aβ-induced microglial activation triggers the release of inflammatory cytokines, including interleukin (IL)-1β and tumor necrosis factor (TNF)-α, further promoting neuroinflammation. This prolonged activation leads to a shift from the homeostatic microglial phenotype to an acute inflammatory phenotype, which, over time, can progress to the DAM/MGnD state, contributing to the advancement of the disease [25,26,27].

Additionally, genetic variations, such as the APOE ε4 allele, play a key role in the onset and progression of AD, with carriers showing faster disease progression. The APOE gene exists in three common forms: ε2, ε3, and ε4. While APOE ε3 is neutral, APOE ε4 increases the risk of AD, with those carrying two copies (homozygous) facing an even higher risk and earlier onset of the disease. APOE ε4 is linked to impaired Aβ clearance in the brain, contributing to amyloid plaque buildup, a hallmark of AD. However, not everyone with APOE ε4 develops AD, highlighting the complex interaction of genetics, environment, and lifestyle in disease progression. Beyond APOE ε4, mutations in APP, PSEN1, and PSEN2 also contribute to AD. Identifying these mutations helps categorize patients into genetic subtypes, essential for personalized treatment strategies, particularly immunotherapies [28,29,30].

Immune responses, including pro-inflammatory markers like IL-1β and TNF-α, as well as regulatory factors such as IL-10, further stratify patients into immune subtypes, informing targeted therapies. Combining genetic markers and immune profiles, including cerebrospinal fluid (CSF) biomarkers, enables tailored immunotherapies, integrating treatments targeting Aβ deposition and tau aggregation [31].

CSF biomarkers such as CSF Aβ42, CSF t-tau, and CSF p-tau aid early AD diagnosis, improving diagnostic accuracy and differentiating AD from other dementias. While amyloid-PET is more precise, CSF biomarkers are a more accessible and cost-effective alternative. Ongoing validation of new biomarkers is critical for advancing AD diagnosis and treatment [32,33]. Personalized treatments should adapt to changes in genetic and immune biomarkers over time. Regular monitoring allows for adjustments in treatment plans and drug dosages, optimizing therapy and minimizing toxicity, especially in patients with mutations in drug-metabolizing genes. Considering both genetic and immune factors can refine immunotherapy approaches, enhancing the relevance and effectiveness of AD treatments tailored to individual patients.

Fecal microbiota transplantation (FMT) has emerged as a promising strategy for AD treatment, demonstrating efficacy in reducing glial responses and cognitive impairments in animal models [34,35]. However, challenges remain, including understanding the mechanisms of FMT, donor selection, and potential side effects [36].

In addition to gut-targeted therapies, immunotherapy represents a promising approach for AD treatment. Immunotherapy involves the use of immune preparations to regulate and modify the host’s immune system to prevent or treat diseases [37]. Immunotherapy can be broadly categorized into two types: active and passive immunotherapy [38]. Active immunotherapy works by stimulating and enhancing the host’s immune response, while passive immunotherapy involves the direct administration of antibodies to the host.

Specifically, immunotherapy for AD holds significant promise as it aims to harness the body’s immune system to target and eliminate the pathological hallmarks of the disease, such as amyloid plaques, tau tangles, and dysfunctional microglia. In this paper, we will explore the role of immune modulation in AD, focusing on the various immunotherapeutic strategies currently under investigation and their potential to alter disease progression and improve patient outcomes. By advancing our understanding of these immunotherapies, we hope to contribute to the evolving landscape of AD treatment.

## 2. Immune System Modifications in AD

### 2.1. Microglia and AD Pathology

Microglia, the intrinsic immune guardians of the central nervous system (CNS), are pivotal in maintaining CNS homeostasis and defending against pathological insults. They intricately regulate neuronal processes during development, including proliferation, survival, synaptic pruning, and plasticity. Moreover, microglia act as vigilant sentinels, continuously surveilling the local environment to respond swiftly to danger signals under pathological conditions. Once activated, they enhance phagocytosis, clear debris and toxins, and coordinate inflammatory responses to restore cerebral homeostasis [39,40,41,42,43].

In AD, microglia exhibit a complex role characterized by diverse phenotypes. Traditionally categorized into pro-inflammatory M1 and neuroprotective M2 types, recent advances challenge this simplistic dichotomy. Transcriptomic and proteomic analyses reveal a spectrum of microglial subtypes in AD, including DAM, PAM and MGnD, each with distinct gene expressions and functions. DAM, for instance, downregulates homeostatic genes while upregulating AD-associated risk genes, whereas MGnD enhances phagocytosis and inflammatory responses to neuronal injury. These phenotypes’ roles in disease progression remain poorly understood despite their significant implications [18,19,20,21,22,23].

In terms of metabolism, the energy requirements of microglia shift as they transition through these phenotypic states. In acute inflammation, microglia rely on glycolysis, akin to the “Warburg effect” observed in tumor cells, which supports a heightened inflammatory response and oxidative stress. On the other hand, homeostatic microglia, when not activated, predominantly rely on oxidative phosphorylation (OXPHOS), coupled with the tricarboxylic acid (TCA) cycle, for ATP production, facilitating maintenance of neuroprotective functions. In the DAM/MGnD state, microglia show altered lipid metabolism, with a pronounced shift toward increased lipid biosynthesis and remodeling, supporting inflammatory processes and the production of pro-inflammatory mediators. This change in lipid metabolism further drives neuroinflammation and the accumulation of neurotoxic factors. Furthermore, DAM/MGnD microglia, distinct from both homeostatic and acute inflammatory microglia, play a significant role in neurodegeneration, not only through pro-inflammatory cytokine release but also by promoting the deposition of toxic protein aggregates, including Aβ and hyperphosphorylated tau.

In contrast, cytokines like IL-4 and IL-10 are involved in maintaining a homeostatic microglial state, promoting an anti-inflammatory environment that enhances Aβ clearance and mitigates tau hyperphosphorylation. During this phase, microglia rely on oxidative phosphorylation, supporting repair processes and reducing overall inflammation [24,25,26].

In AD pathology, microglial activation precedes Aβ plaque accumulation and fluctuates bimodally during disease progression. Initially, microglia activation increases, exerting a neuroprotective effect by clearing Aβ. However, as the disease advances, microglia transition to a pro-inflammatory phenotype, exacerbating neuroinflammation and neuronal damage. Clinical studies corroborate the correlation among microglia activation, Aβ, and tau pathology in AD patients, emphasizing their pivotal role in AD pathogenesis [18].

Deciphering the interrelationship between microglial activation and AD pathology is crucial for understanding disease mechanisms and developing effective therapies. By unraveling the complexities of microglial phenotypes and their metabolic alterations, researchers aim to modulate microglial function to mitigate neuroinflammation and promote neuroprotection in AD. This multidimensional approach holds promise for therapeutic interventions targeting microglia to combat AD progression.

#### 2.1.1. Microglia and Aβ

Reactive microglia exhibit a significant association with amyloid plaques in AD brains. Aβ acts as a trigger for microglial activation, stimulating the release of inflammatory cytokines such as IL-1β and TNF-α, thereby fostering microglia-mediated neuroinflammation and neuronal death. Notably, smaller Aβ oligomers are particularly potent in inducing microglial inflammatory responses. Despite this, activated microglia in the early stages of AD demonstrate enhanced phagocytic capacity, facilitating Aβ clearance and forming barriers around amyloid plaques to impede their growth. However, chronic microglial activation in advanced stages compromises phagocytic ability due to downregulated Aβ phagocytic receptors, leading to ineffective Aβ clearance and maladaptive immune responses. Furthermore, microglia exacerbate Aβ plaque spread by activating NLRP3 inflammasomes, promoting the formation of apoptosis-associated speck-like protein containing a CARD (ASC)-Aβ aggregates and amplifying neuroinflammation [44,45].

In the context of AD pathogenesis, Aβ accumulation profoundly influences microglial responses. Aβ induces microglial phagocytosis, cytokine release, and proliferation. Indeed, Aβ interacts with multiple receptors on microglial surfaces, such as SR-A1, TLR4, TREM2, RAGE, P2X7, and FPRL1 receptors, facilitating adhesion and subsequent phagocytosis. Moreover, Aβ triggers microglial secretion of pro-inflammatory cytokines and neurotoxic substances, including TNF-α, IL-1, IL-6, IFN-γ, and reactive oxygen species, contributing to neuroinflammation and neuronal damage. For instance, TNF-α directly enhances the production of Aβ and boosts the expression of β- and γ-secretase, leading to increased Aβ buildup. Likewise, IL-1β encourages Aβ accumulation by prompting astrocytes to release α1-antichymotrypsin (ACT) and influencing the synthesis and processing of APP. Other microglial inflammatory mediators such as IL-6 and IFN-γ also contribute to AD progression, impairing cognitive function and perpetuating neuroinflammation [46,47].

The interaction between Aβ and microglia accelerates inflammatory responses and neural damage, perpetuating AD development. Understanding the complex interplay between Aβ and microglia is crucial for unraveling the mechanisms underlying AD pathogenesis and developing effective therapeutic strategies.

#### 2.1.2. Microglia and Tau

The amyloid cascade–inflammation hypothesis suggests that the activation of microglia by Aβ can trigger inflammatory reactions that ultimately lead to the aggregation of tau protein. Tau, primarily found in neurons, plays a role in regulating microtubule assembly and stability. In AD, tau undergoes various post-translational modifications, notably hyperphosphorylation, leading to its disassociation from microtubules, aggregation, and accumulation within neurons, ultimately impairing neuronal function and causing synaptic dysfunction (Figure 1) [48,49]. In mouse models of tauopathies, immunosuppressants have been shown to dampen inflammatory responses, alleviate tau pathology, and extend lifespan. Microglia are key drivers of tau pathology and are implicated in the propagation of pathological tau. Although microglia can phagocytose tau, hyperphosphorylated tau disrupts the interaction between neurons and microglia. The CX3CL1/CX3CR1 axis, crucial for various brain functions including synaptic integration and emotional behavior, is involved in tau phagocytosis by microglia. While tau binding to CX3CR1 triggers tau phagocytosis, phosphorylation of tau at the S396 site reduces its binding affinity to CX3CR1. In late-stage AD, dysfunction in the CX3CL1/CX3CR1 axis may diminish microglial phagocytic activity. Elevated levels of CX3CL1 in the cerebrospinal fluid and blood of AD patients suggest its potential as a blood biomarker, particularly in early-stage diagnosis [50,51].

In mice models, the CX3CR1 receptor, crucial for microglial function, facilitated the phagocytosis of tau, and its absence hampered tau uptake, resulting in increased tau hyperphosphorylation. Additionally, activated microglia can aid in the spread of tau through processes like phagocytosis and exosome secretion [52,53,54,55]. The aggregation of pathogenic tau further activates microglia and triggers the release of inflammatory cytokines, which in turn promote tau hyperphosphorylation via the p38 MAPK pathway [56]. Tau itself can also activate the NLRP3 inflammasome, leading to the release of inflammatory molecules like IL-1β and IL-18 through an ASC-dependent pathway. Deficiencies in NLRP3 or ASC partially reduce tau hyperphosphorylation by influencing tau kinases in an IL-1β-dependent manner [54,57,58,59].

Overall, the interaction between Aβ-induced microglial activation and tau pathology plays a significant role in the progression of AD. Understanding these mechanisms offers insights into potential therapeutic targets for mitigating neuroinflammation and tau aggregation in AD.

### 2.2. Peripheral Blood Immune Cells

Neuroinflammation and immune-related mechanisms are crucial in the pathophysiology and progression of AD [60]. The interaction between peripheral immune cells and the CNS occurs through three primary pathways: (1) the blood–brain barrier (BBB), which connects the brain to the circulation [61]; (2) the choroid plexus (CP), which forms an interface between the blood and cerebrospinal fluid (CSF) [62]; and (3) the meninges, an immune–blood–brain interface that allows immune cells to bypass the BBB and enter directly into the brain via specialized skull bone marrow channels [63].

Myeloid cells, which are part of the innate immune system, play a key role in neuroinflammation and neurodegeneration [64]. These cells include peripheral immune cells such as neutrophils and monocytes, which can infiltrate the brain and exacerbate peripheral inflammation by releasing pro-inflammatory cytokines like interleukin-1β (IL-1β) and interleukin-6 (IL-6). Lymphoid cells, such as T and B cells, are involved in adaptive immunity. While their role in AD is still not fully understood, recent studies indicate that they can infiltrate the brain [65].

The integrity of the BBB is compromised by the accumulation of Aβ and tau in the AD brain and blood vessels, resulting in the release of pro-inflammatory mediators and the infiltration of myeloid cells into the brain [61]. Furthermore, Pietronigro et al. demonstrated that neutrophil-specific protease cathepsin G accumulates in the brain and blood vessels of AD patients [66], while Zenaro et al. confirmed the infiltration of neutrophils into the brains of AD mice and observed that neutrophil extracellular traps (NETs) promoted amyloid plaque formation and tau tangles, leading to worsened cognitive decline [67]. El Khoury et al. highlighted that blood monocytes enter the brain through the vasculature, migrating toward and associating with amyloid plaques [68].

In human peripheral blood, platelets emerge as the primary source of Aβ peptides, with more than 90% of circulating Aβ being derived from platelets. Additionally, the liver contributes to the peripheral clearance of circulating Aβ, influencing peripheral blood Aβ levels indirectly [69]. Notably, the liver’s involvement in Aβ metabolism has been underscored in AD pathology, with studies showing alterations in liver function and Aβ clearance in AD transgenic mice [70]. Although other peripheral sources of Aβ, such as endothelial vascular cells and skeletal muscle, exist, they are not the main contributors to peripheral blood Aβ levels [71,72,73].

The impact of peripheral Aβ on peripheral innate immune cells, including neutrophils, monocytes, macrophages, and natural killer (NK) cells, remains an area of active investigation. Pathological changes in both central and peripheral immune responses throughout the AD process have been observed. Understanding how peripheral Aβ influences peripheral innate immune cells in AD is crucial for elucidating their causative roles, whether protective or harmful. This understanding may inform the development of therapeutic strategies targeting the immune system at different stages of AD [74].

#### 2.2.1. Monocytes

The role of monocytes in AD is multifaceted and complex, with implications for disease progression and potential therapeutic interventions. Peripheral monocytes, derived from hematopoietic stem cells, encompass various subpopulations with distinct functions and phenotypes. Classical monocytes are involved in phagocytosis and immune responses, while non-classical monocytes excel in complement and Fc-mediated adhesion and phagocytosis. Intermediate monocytes play a crucial role in antigen presentation and cytokine secretion. During the course of AD, classical monocytes decrease progressively, while the percentages of non-classical and intermediate monocytes increase with the Aβ load [75,76,77].

The levels of C-reactive protein (CRP) and complement system components derived from monocytes fluctuate during AD progression, reflecting a shift from an anti-inflammatory to a pro-inflammatory phenotype. Although CC-chemokine ligand 2 (CCL2) facilitates the recruitment of monocytes to the central nervous system (CNS) in response to Aβ stimulation, recent evidence challenges the notion that infiltrating monocytes significantly contribute to the brain’s myeloid population. Studies such as those by Reed-Geaghan et al. (2020) [78] and Wang et al. (2016) [79] demonstrate that plaque-associated myeloid cells in AD are derived predominantly from brain-resident microglia, with minimal contribution from circulating monocytes. Similarly, Naert and Rivest (2013) [80] highlighted that resident microglia, rather than infiltrating CCR2+ monocytes, primarily mediate CNS immune responses under both normal and AD conditions. Naert and Rivest also emphasized that while CCR2+ monocytes can infiltrate the CNS in certain inflammatory contexts, such as multiple sclerosis or severe BBB disruption, their contribution to the brain’s myeloid population in AD is minimal.

While earlier research suggested that infiltrating monocytes could regulate brain inflammation, protect cortical neuron synapses, and aid in Aβ clearance by internalizing and transporting it into the bloodstream, these roles appear to be primarily mediated by resident microglia under normal conditions. Additionally, studies using CCR2-deficient mouse models have shown that the absence of CCR2 does not significantly impact amyloid plaque burden, further supporting the limited involvement of monocytes in CNS parenchymal processes in AD. Furthermore, the phagocytic capacity of monocytes is known to decline in AD, largely due to reductions in adhesion molecules and Aβ internalization-related receptors, which may further limit their potential involvement in Aβ clearance. These findings collectively underscore the dominant role of resident microglia in AD-related immune responses and the comparatively limited contribution of infiltrating monocytes to CNS pathology [70,71,72,73,74,78,79].

Monocytes can also infiltrate the meninges under both steady-state and inflammatory conditions, with skull bone marrow serving as a key reservoir for monocyte precursors. These cells, upon entering the dura via specialized vascular channels, can differentiate into antigen-presenting cells, such as dendritic cells, or into macrophages. This meningeal trafficking provides additional immune surveillance and modulation, particularly in disease states associated with neuroinflammation. Enhanced monocyte recruitment to the meninges has been observed in models of neuroinflammation, highlighting the potential for meningeal immune responses to influence AD pathology [81].

Monocytes can differentiate into monocyte-derived macrophages (MDMs), which infiltrate the brain during the late stages of Aβ pathology. MDMs exhibit a robust capacity for Aβ removal by upregulating phagocytic receptors. Additionally, MDMs can adopt either pro-inflammatory (M1) or anti-inflammatory (M2) phenotypes depending on the local microenvironment. Changes in the proportion of M1 and M2 macrophage subsets have been observed in AD patients, with implications for cognitive function [82].

Recent studies have highlighted the therapeutic potential of modulating monocyte function in AD. Strategies to improve energy metabolism in monocytes can enhance Aβ phagocytosis and reduce brain Aβ deposition and neuroinflammation, ultimately improving cognitive function. Moreover, the absence of specific monocyte subsets, such as Ly6C monocytes, can exacerbate Aβ deposition in the brain. Understanding the molecular mechanisms underlying monocyte dysfunction in AD, such as the role of cytokines and signaling pathways, may lead to novel therapeutic targets [83,84].

#### 2.2.2. Neutrophils

Neutrophils, as the most abundant peripheral immune cells, play a complex role in AD pathology. While traditionally known for their role in combating pathogens and producing cytokines, recent research has highlighted their involvement in AD-related neuroinflammation and pathology [85,86].

In AD patients, the number of neutrophils significantly increases, particularly when memory deficits appear. Neutrophils express integrin lymphocyte function-associated antigen 1 (LFA-1) on their surface, facilitating their infiltration into the brain and attachment to cerebral blood vessels. When exposed to soluble oligomeric Aβ, neutrophils interact with intercellular adhesion molecule-1 (ICAM-1) on vessels via LFA-1 integrin, leading to the blockade of cerebral blood flow and subsequent Aβ production, tau hyperphosphorylation, and neurodegeneration. Conversely, depletion of neutrophils improves cerebral blood flow and mitigates AD-related neuropathology in mouse models [67,87].

Infiltrating neutrophils also stimulate the production of inflammatory cytokines and the formation of neutrophil extracellular traps (NETs), promoting microglial activation and neuroinflammation. Activated microglia, in turn, promote the recruitment of neutrophils to the brain through the secretion of inflammatory cytokines. Furthermore, neutrophil-related proteins are elevated in AD patients and positively correlated with disease progression [88].

However, the role of neutrophils in AD pathology is not entirely detrimental. Some studies have suggested a subset of neutrophils that contribute to neuronal survival in the central nervous system (CNS). Neutrophils may play dual roles in the pathogenesis of AD, with both harmful and potentially beneficial effects.

#### 2.2.3. NK Cells

NK cells, a subset of innate lymphoid cells, have emerged as key players in the complex pathogenesis of AD. Despite a stable frequency in circulation, NK cells exhibit dynamic changes in cytotoxic activity during AD progression. Studies reveal heightened expression of granzyme B and inflammatory cytokines in NK cells of individuals with amnestic mild cognitive impairment (MCI) and mild AD; however, findings in severe AD are inconclusive [89,90]. This suggests a potential adverse role of NK cells in cognitive function. Furthermore, in mouse models, NK cells exacerbate neuroinflammation and cognitive decline, while anti-NK cell treatment improves cognitive function and mitigates neuroinflammation, indicating a detrimental impact of NK cells in AD [91].

However, the role of NK cells in AD pathology remains controversial. Some studies report lowered cytotoxic function in AD patients compared to healthy individuals, while others show enhanced cytotoxic capacity in AD patients. NK cells may contribute to chronic inflammation in the central nervous system (CNS) of AD by stimulating macrophages [90,92]. Additionally, NK cells may participate in clearing Aβ plaques, as evidenced by mouse model studies. While NK cell infiltration into the brains of AD mouse models has been observed, research on NK cell infiltration in AD patients is limited. Single-cell RNA sequencing data suggest peripheral NK cells may infiltrate the brain in AD patients, contributing to neuroinflammation, but further studies are needed to confirm these findings [93,94].

In individuals with mild AD, NK cell activation capacity appears unaltered, as evidenced by consistent expression of CD107a, granzyme B, and IFN-γ. However, the status of NK cells in severe AD remains elusive. Mouse models lacking NK cells, alongside T and B cells, exhibit elevated Aβ levels, indicating a potential role for NK cells in regulating Aβ deposition. Interestingly, these mice also display altered microglial morphology, suggesting NK cells’ involvement in microglial activation. Yet, further investigations are warranted to discern whether these effects directly result from NK cell deficiency or the absence of other lymphocyte subpopulations [89,95].

Contrary to expectations, T cell- and B cell-deficient mice, albeit possessing functional NK cells, exhibit reduced Aβ plaque burden. Nonetheless, unlike models lacking NK cells, their microglial activation remains unchanged, indicating a differential impact on microglial function. These findings suggest a nuanced role for NK cells in AD pathology, potentially influencing microglial activation pathways distinct from T and B cells [95].

In conclusion, NK cells emerge as early responders implicated in the pathological changes observed in AD. Their dynamic cytotoxic activity, inflammatory cytokine release, and potential role in Aβ plaque clearance highlight their complex involvement in AD pathology. Further research is needed to elucidate the specific mechanisms underlying NK cell function in AD and its potential as a therapeutic target.

### 2.3. Adaptive Immunity

Recent research has highlighted the involvement of the adaptive immune system in AD; however, its exact role remains unclear compared to the innate immune system. T and B cells, crucial components of the adaptive immune system, exhibit receptors capable of recognizing diverse antigens and maintaining immunological memory and self-tolerance. Studies indicate an increase in T cells, particularly CD8+ T cells, in regions like the cerebrospinal fluid (CSF), leptomeninges, and hippocampus of AD patients and mouse models, correlating with severe pathology. Research demonstrates both beneficial and detrimental effects of adaptive immunity in AD pathogenesis, with genetic ablation of immune cell populations accelerating amyloid pathology, while interventions like immunoglobulin G (IgG) replacement or bone marrow transplantation reduce amyloid deposition. While the role of adaptive immunity in AD is significant, questions remain about the temporal dynamics of immune changes, specific lymphocyte involvement, and the interplay between adaptive and innate immunity across AD stages, critical for developing targeted therapeutic strategies [96,97].

#### 2.3.1. T Cells

In the context of AD, the infiltration of peripheral T cells into the CNS is not solely dependent on disruptions in the blood–brain barrier (BBB). T cells are capable of crossing an intact BBB through mechanisms involving cytokines, chemokines, and their receptors, which facilitate the migration of T cells across brain endothelial cells. Elevated levels of chemokine receptors such as CXCR2, CCR5, and MIP-1α facilitate T cell movement through the tight junctions of the BBB, particularly in the presence of Aβ accumulation [98,99].

Once T cells infiltrate the brain parenchyma, they interact with various cell types, including microglia, astrocytes, and neurons. CD4+ T cells, also known as helper T cells, exhibit diverse functions depending on their subtype. Effector T cells (Teff) secrete pro-inflammatory cytokines such as IFN-γ, IL-17, and TNF-α, contributing to neuroinflammation and neuronal damage. Conversely, regulatory T cells (Treg) exert anti-inflammatory effects by secreting immunosuppressive cytokines and promoting neuroprotection [100,101].

The balance between pro-inflammatory Teff and anti-inflammatory Treg is crucial for maintaining immune homeostasis in the brain. During the early stages of AD, Treg infiltration confers neuroprotection and suppresses disease progression by modulating microglial activation and promoting Aβ clearance. However, as the disease advances, the immunosuppressive capacity of Treg may diminish, leading to dysregulated neuroinflammation and cognitive decline [102].

CD8+ T cells, also known as cytotoxic T cells, play a significant role in AD neuropathology by directly targeting neuronal cells and promoting neurodegeneration. These cells express cytotoxic molecules such as perforin and granzyme, which induce neuronal apoptosis and compromise neuronal integrity. CD8+ T cells can also disrupt synaptic plasticity and impair cognitive function by secreting pro-inflammatory cytokines and compromising BBB integrity [103].

Furthermore, the interaction between CD8+ T cells and myeloid cells, such as microglia, exacerbates neuroinflammation and neuronal damage in AD. The presence of tissue-resident memory CD8+ T cells in the brain further contributes to neuroinflammatory responses and disease progression [98].

This dysregulation of T cell-mediated immune responses in AD underscores the complex interplay between the adaptive immune system and neurodegenerative processes. Understanding the roles of different T cell subtypes and their interactions with other immune and neuronal cells is essential for developing targeted immunotherapies and therapeutic interventions for AD.

#### 2.3.2. B Cells

B cells play a crucial role in the adaptive immune response, contributing to both humoral immunity through immunoglobulin secretion and antigen presentation to T cells, thereby initiating T cell-mediated adaptive responses. In AD, alterations in B cell function and population dynamics have been observed. While the frequency of IgG-producing B cells is increased in the serum of AD patients, the total number of B cells decreases [104,105,106,107].

Notably, B cells in AD patients express various neurodegenerative-related proteins and engage in extensive crosstalk with inflammation-related proteins. The upregulation of pro-inflammatory receptors on B cells, such as CCR6 and CCR7, has been noted in moderate to severe AD cases. Additionally, the presence of misfolded Aβ protein in AD can stimulate the generation of Aβ-targeting autoantibodies by B cells. These autoantibodies have been shown to mitigate the toxicity of Aβ oligomers and protofibrils, leading to improvements in spatial memory in AD transgenic mouse models [106,108].

Studies in AD mouse models have further elucidated the role of B cells in amyloid plaque clearance. Intracerebral administration of anti-Aβ antibodies or pre-immunized mouse IgG has been found to reduce the number of Aβ plaques by enhancing microglial phagocytic activity. Conversely, immune cell ablation in AD mice exacerbates amyloid pathology, partially due to the absence of B cell-produced IgG and impaired microglia-mediated phagocytosis [109].

However, conflicting results have emerged regarding the impact of B cell depletion on tau pathology in AD. While tau-specific antibodies are detected in the brain and serum of AD individuals, studies in tauopathy mice suggest that B cells do not influence tau pathology. Importantly, many researchers believe these conflicting results are influenced by differences in experimental models, including variations in genetic backgrounds, methods of B cell depletion, and the specific tauopathy models used. For instance, genetic B cell depletion in tau transgenic mice has been shown to moderately exacerbate spatial memory deficits [110].

The precise role of B cells in AD etiology and progression remains unclear, with conflicting findings necessitating further investigation. Understanding the nuanced interactions among B cells, other immune cells, and neuronal components in the context of AD pathology is crucial for developing targeted therapeutic strategies.

### 2.4. Immune Microenvironment in the Brain Parenchyma

The interplay between innate and adaptive immunity in AD is complex. Under homeostatic conditions, Treg and effector T cells engage with microglia, releasing anti- or pro-inflammatory cytokines like IL-10, TGF-β, IL-4, and IFN-γ. Brain-resident cells in neurodegenerative or aged niches, such as neurons, microglia, astrocytes, endothelial cells, or macrophages, produce cytokines, neurotoxic reactive oxygen species (ROS), and inducible nitric oxide synthase (iNOS), initiating a neuroinflammatory cascade in the brain parenchyma that can attract T cell invasion [111,112,113].

The changing brain milieu, including alterations to brain barriers like the meninges, BBB, and blood–CSF barrier, can facilitate T cell infiltration due to increased expression of adhesion molecules and integrins. Activated T cells secrete pro-inflammatory and neurotoxic mediators, perpetuating the inflammatory cascade and inducing microglial state transition, potentially initiating or accelerating brain atrophy [114].

A subset of microglia, particularly prevalent in the presence of tau pathology, exhibits high expression of CD80, CD86, MHC-I, and MHC-II complex genes. Whether this “dendritic-cell-like” microglial population plays a role in antigen presentation to T cells in AD-like disease states and their interaction with T cell subsets requires further investigation [115].

Regional brain atrophy progression in AD strongly correlates with tau accumulation rather than amyloid deposition. The involvement of T cells in neurodegeneration in AD model systems with AD-type pathologies remains uncertain. If T cells are implicated in mediating neurodegeneration, they could potentially serve as modulatory targets for AD treatment. Immune checkpoint blockade targeting the PD-1/PDL1 pathway shows promise in modifying factors implicated in AD and dementia.

T cells predominantly release IFNs, particularly IFN-γ, which, upon local infiltration and cytokine release, facilitate strong intercellular communication between resident innate and adaptive immunity, playing critical roles in disease states. Further studies are needed to determine if IFNs, particularly IFN-γ signaling, could be viable therapeutic targets [116].

Exploring the role of adaptive immunity in tauopathy and neurodegeneration, along with the immune microenvironment in brain parenchyma, is essential for detecting direct evidence of whether adaptive immunity influences neurodegeneration.

## 3. Immunotherapy for AD

Currently, there is no cure for AD. In fact, approximately 200 unsuccessful clinical trials have been conducted in the past decade [117]. Immunotherapy, inherently linked to immunology, aims to provide treatment strategies for diseases in which immune responses are altered and can potentially be modulated [38,118].

Immunotherapy is broadly classified into active and passive types. Active immunotherapy involves stimulating and enhancing the host’s immune response, while passive immunotherapy entails the direct administration of antibodies to the host [38].

Vaccines play a significant role in immunotherapy, particularly in preventing and managing infectious diseases, due to their demonstrated effectiveness. However, passive immunotherapies are often more suitable for older patients, as they tend to respond less effectively to active immunotherapy [119]. One major limitation of passive immunization is the need for repeated dosing, which can increase treatment costs. In contrast, active immunotherapies typically require fewer injections, making them more cost-effective. In the context of AD, key targets for immunotherapy include Aβ plaques (Figure 2), tau proteins, and microglia. This section will discuss current active and passive immunotherapy approaches focusing on these primary components of AD.

### 3.1. Immunotherapy Based on Targeting Aβ

Immunotherapy for AD targeting Aβ focuses on reducing the toxic effects of Aβ plaques in the brain, which are strongly associated with the cognitive decline seen in AD. Since Aβ is thought to play a crucial role in the disease’s progression, researchers have developed different strategies to promote its clearance or prevent its accumulation. These strategies are broadly classified into two approaches: active and passive immunotherapy. Various therapeutic agents are designed to target different forms of Aβ aggregates, such as monomers, oligomers, protofibrils, fibrils, and plaques, with the goal of preventing their formation, enhancing their removal, or neutralizing their harmful effects (Figure 3).

Despite significant efforts, many therapeutic approaches targeting Aβ have faced challenges, highlighting the complexity of intervening in amyloid pathways. Gamma-secretase and beta-secretase inhibitors, which aim to prevent Aβ production, have largely failed in clinical trials. For example, gamma-secretase inhibitors like semagacestat disrupted Notch signaling, leading to severe side effects, including cognitive decline and peripheral toxicity, due to the enzyme’s role in processing multiple essential substrates. Moreover, inconsistent pharmacokinetics and transient inhibition paradoxically increased Aβ production in some cases due to oscillatory effects. Beta-secretase 1 (BACE1) inhibitors, such as Merck’s verubecestat, also demonstrated limited efficacy. Even with near-maximal reductions in cerebrospinal fluid Aβ and modest decreases in brain amyloid load, verubecestat failed to slow disease progression in mild-to-moderate AD. Trials of atabecestat in asymptomatic individuals were similarly discontinued due to safety concerns, including elevated liver enzyme levels. These failures have not only tempered expectations for amyloid-centric strategies but also raised questions about the amyloid hypothesis itself, suggesting that Aβ-independent mechanisms may drive disease progression in advanced stages [120,121].

Monoclonal antibody therapies, on the other hand, have shown greater promise in targeting Aβ. These therapies are designed to selectively bind to toxic aggregated forms of Aβ, such as plaques and oligomers, thereby reducing their pathological burden without interfering with critical physiological pathways. For instance, aducanumab and other antibodies have demonstrated potential in slowing cognitive decline, particularly when administered early in the disease course. These therapies underscore the importance of precise targeting and timing of intervention to maximize therapeutic efficacy.

In the following sections, different types of active and passive immunotherapies based on Aβ will be explored in greater detail, highlighting their mechanisms of action, clinical outcomes, and potential future directions.

#### 3.1.1. Active Immunotherapy

Active immunotherapy typically involves the stimulation of patient’s immune system to produce antibodies that specifically target Aβ, while passive immunotherapy administers pre-formed antibodies or other biologics directly to the patient. Both approaches aim to reduce the burden of toxic Aβ species, potentially slowing the disease’s progression and improving cognitive function [122]. Various vaccines have been developed with the goal of reducing Aβ plaque buildup in the brain and slowing cognitive decline. However, the journey has been challenging, with mixed results and some setbacks. Below is an overview of three key active immunotherapy candidates: AN1792, amilomotide (CAD106), and UB-311.

AN1792 was the first anti-Aβ vaccine tested in human clinical trials. This vaccine consisted of a synthetic full-length Aβ42 peptide combined with the QS-21 adjuvant, which enhances the immune response. In the phase IIa clinical trial (NCT00021723) (ClinicalTrials.gov), around 19.7% of the patients treated with AN1792 were identified as antibody responders, meaning they produced high levels of anti-Aβ antibodies. For these individuals, the vaccine appeared to reduce Aβ deposition in the brain, improve neuropsychological test scores, and lower cerebrospinal fluid (CSF) tau levels, a marker of neuronal damage. However, these positive effects were only seen in the antibody responders, suggesting limited overall efficacy. A major issue that emerged was that 6% of the participants developed T cell-mediated meningoencephalitis, a severe brain inflammation. This complication led to the early termination of the trial (ClinicalTrials.gov) [123,124]. A follow-up study later showed that some of the antibody responders maintained low but detectable levels of anti-Aβ antibodies, which correlated with slower cognitive decline and some long-term functional benefits. Despite these results, the risks associated with AN1792 halted further development [125].

In the wake of AN1792’s failure, researchers focused on creating safer vaccines. One of these is amilomotide (also known as CAD106), which uses only the N-terminal fragment of Aβ (Aβ1-6) as a B-cell epitope, rather than the full Aβ peptide. This strategy was designed to generate anti-Aβ antibodies without triggering a T-cell response, thus reducing the risk of meningoencephalitis [126]. In phase I trials (NCT00411580), CAD106 demonstrated a good safety profile and an adequate antibody response. Subsequent phase II trials (NCT00733863, NCT00795418, NCT00956410, NCT01023685, and NCT01097096) confirmed that CAD106 struck a balance between inducing an immune response and maintaining tolerability. However, as the vaccine entered a larger phase II/III trial (NCT02565511), unforeseen issues arose. Participants experienced unexpected changes in cognitive function, brain volume, and body weight, which led to the early termination of the trial. These adverse effects indicated that more refinement was needed to optimize the vaccine’s safety and efficacy (7 clinical trials).

UB-311 represents another approach to active immunotherapy. This vaccine is composed of two synthetic Aβ1-14 peptides that act as B-cell epitopes, each linked to different helper T-cell epitopes via a proprietary delivery system (UBITh^®^, United Biomedical, Inc. UBI, New York, NY, USA). This system is designed to enhance the immune response while reducing the risk of harmful T-cell inflammation [127]. In a phase II trial (NCT02551809), UB-311 showed promise, achieving a 100% responder rate with strong immunogenicity, meaning that all participants produced anti-Aβ antibodies. Preliminary results suggested cognitive improvements in patients with early to mild AD. However, the trial had some limitations, including the lack of a placebo group for comparison. The efficacy of the vaccine was inferred from a subgroup analysis comparing mild and moderate AD patients based on their AD Assessment Scale-Cognitive Subscale (ADAS-Cog) scores. Another phase II trial (NCT03531710) was halted due to a treatment assignment error, which limited the further evaluation of UB-311’s potential [127] (two clinical trials).

These vaccines demonstrate the potential of active immunotherapy in AD, but they also highlight the complexities involved in developing safe and effective treatments. While AN1792 showed initial promise, the risks of meningoencephalitis ended its development. Subsequent vaccines like amilomotide and UB-311 have sought to reduce the likelihood of such side effects, but challenges remain, particularly in balancing immune activation with safety and managing unpredictable cognitive effects. Further research and refinement of these approaches are needed to fully harness the potential of active immunotherapy for AD.

#### 3.1.2. Passive Immunotherapy

The landscape of passive immunotherapy for AD has evolved significantly, particularly with the advent of several mAbs targeting Aβ. These therapies aim to mitigate the fundamental pathological processes associated with AD by facilitating the clearance of Aβ aggregates, which include soluble oligomers and insoluble fibrils [128].

Aducanumab (BIIB037) is a human IgG1 monoclonal antibody that specifically binds to the N-terminus of Aβ in its extended conformation. It is designed to target both Aβ aggregates and plaques [129]. In the phase Ib randomized trial, PRIME (NCT01677572), aducanumab treatment led to significant reductions in amyloid PET standard uptake value ratio (SUVR) scores, particularly in patients receiving the higher dose of 10 mg/kg at 54 weeks. The treatment was associated with a decrease in brain amyloid burden in a dose- and time-dependent manner among individuals with prodromal or mild AD. Improvements were also observed in cognitive measures, as indicated by the Clinical Dementia Rating-Sum of Boxes (CDR-SB) and Mini-Mental State Examination (MMSE) scores, suggesting a potential delay in clinical progression [130]. However, aducanumab was linked to amyloid-related imaging abnormalities (ARIA), specifically vasogenic edema (ARIA-E), occurring in a dose-dependent manner in 3–41% of recipients, especially in carriers of the APOE ε4 allele. Although a phase II study was bypassed due to promising phase I results, two identically designed phase III trials, ENGAGE (NCT02477800) and EMERGE (NCT02484547), were terminated in March 2019 after a futility analysis suggested a low probability of achieving statistical significance [131]. This decision was based on the assumption of consistent treatment effects across the trials and over time, which Biogen later argued were invalid.

Subsequent analyses revealed discordant results between the trials. EMERGE met its primary endpoint, showing a 22% reduction in clinical decline in the high-dose group, while ENGAGE failed to demonstrate any clinical benefit. This inconsistency raises significant concerns about the reliability and replicability of the findings. Instead of addressing these uncertainties through a new phase III trial, Biogen pursued regulatory approval using post hoc analyses. These analyses deviated from the prespecified statistical analysis plan, failing to account for multiplicity and Type I error control. Only the high-dose group in EMERGE achieved nominal significance on the primary endpoint (*p* = 0.012), while other doses and outcomes did not meet statistical thresholds. Despite this, Biogen characterized the results as “clinically meaningful”, even though the observed effect size on the CDR-SB scale (−0.39) was modest and inconsistent with ENGAGE.

Biogen’s reliance on amyloid plaque reduction as a surrogate endpoint further complicates the case for aducanumab. Accelerated approval was granted based on evidence of significant plaque reduction, deemed “reasonably likely” to predict clinical benefit. However, the link between amyloid clearance and cognitive improvement remains tenuous. For example, high-dose treatment in ENGAGE showed substantial plaque reduction but cognitive decline, undermining the validity of amyloid reduction as a surrogate marker.

The FDA’s decision to approve aducanumab has faced substantial criticism for prioritizing biomarker outcomes over demonstrated clinical benefits. This approach risks setting a troubling precedent and highlights the need for rigorous trial designs, adherence to prespecified analyses, and greater transparency in regulatory processes to ensure patient safety and therapeutic efficacy [132,133].

Donanemab (LY3002813) is another humanized monoclonal IgG1 antibody that targets the N-terminal pyroglutamate Aβ epitope found primarily in deposited Aβ [134]. In the phase II trial TRAILBLAZERALZ (NCT03367403), donanemab was associated with a reduction in the AD rating scale scores, indicating a smaller decline in cognitive and functional abilities among patients with early-stage AD. Notably, a significant reduction in amyloid plaques was observed, with 54.7% of participants achieving an amyloid-negative status at 52 weeks. The reduction in amyloid burden correlated with less cognitive decline and a decrease in tau progression. Rapid decreases in plasma p-tau217, a biomarker indicative of AD pathology, were noted within 12 weeks of treatment. However, donanemab was also linked to a higher incidence of ARIA-E compared to placebo (26.7% vs. 0.8%), underscoring the need for larger, longer trials to thoroughly evaluate its safety and efficacy [135]. Overall, donanemab demonstrated promising effects in reducing amyloid plaques and slowing cognitive decline, although safety concerns related to ARIA-E remain significant. Ongoing studies, including TRAILBLAZER-ALZ 2 (NCT04437511) and TRAILBLAZER-ALZ 3 (NCT05026866), aim to assess the potential of donanemab in preventing clinical progression in patients with preclinical AD (3 clinical trials).

Lecanemab (BAN2401) is a humanized IgG1 monoclonal antibody that preferentially targets soluble aggregated Aβ, acting on oligomers, protofibrils, and insoluble fibrils. In the phase II trial BAN2401-G000-201 (NCT01767311), although the primary endpoint at 12 months was not met, a reduction in brain amyloid plaques was observed, along with evidence of sustained clinical remission at the highest dose of 10 mg/kg biweekly. Interestingly, while the difference in CDR-SB decline was not significant at 18 months, it was significant at 12 months. The treatment demonstrated a dose-dependent effect, particularly among APOE4 carriers, leading to increased Aβ42 levels and decreased p-tau levels in cerebrospinal fluid (CSF). Although ARIA-E incidence was noted at 9.9%, with higher rates in APOE4 carriers (14.3%), lecanemab was generally well tolerated [136]. These findings highlight lecanemab’s ability to reduce amyloid burden and stabilize cognitive performance, particularly in APOE4 carriers, despite safety concerns related to ARIA-E. Ongoing phase III studies, including Clarity AD (NCT03887455) and AHEAD 3-45 (NCT04468659), are exploring its long-term efficacy and safety in early-stage and preclinical AD patients (2 clinical trials).

Solanezumab (LY2062430), a humanized monoclonal antibody targeting the mid-domain of the Aβ peptide, was subjected to two phase III trials, EXPEDITION 1 (NCT00905372) and EXPEDITION 2 (NCT00904683), which ultimately failed to demonstrate efficacy in slowing cognitive decline in mild-to-moderate AD patients [137]. In another trial, DIAN-TU (NCT01760005) evaluated the effects of solanezumab in patients with dominantly inherited AD; the treatment successfully targeted Aβ but did not lead to cognitive improvement. In fact, it showed a slight worsening of cognitive impairment compared to the control group [138]. Moreover, the current A4 trial (NCT02008357) evaluated solanezumab’s effects in asymptomatic patients with amyloid plaques. In this phase III trial, solanezumab, targeting monomeric amyloid, was tested in 1169 individuals aged 65–85 with preclinical AD, no cognitive impairment, and elevated brain amyloid levels. Participants were randomized to receive solanezumab (up to 1600 mg every 4 weeks) or placebo, with the primary endpoint being the change in the Preclinical Alzheimer Cognitive Composite (PACC) score over 240 weeks. The results showed no significant difference in cognitive decline between groups (*p* = 0.26), with a mean PACC score decline of −1.43 in the solanezumab group and −1.13 in the placebo group. Although amyloid PET levels increased more slowly with solanezumab, the treatment did not slow cognitive decline, indicating a lack of efficacy in preclinical Alzheimer’s disease [139]. These results suggest that solanezumab effectively targets amyloid but lacks significant clinical benefits, raising doubts about its utility in both symptomatic and preclinical AD. Additional trials, including an extension study (EXPEDITION EXT, NCT01127633) and others, also failed to meet primary endpoints (four clinical trials).

Crenezumab (RG7412), another humanized IgG1 monoclonal antibody, targets multiple Aβ forms, including monomers and aggregates, with a particular affinity for oligomers [140]. The phase III trials CREAD (NCT02670083) and CREAD2 (NCT03114657) were halted due to interim analyses indicating low likelihood of meeting primary endpoints. In addition, a phase II, double-blind, placebo-controlled trial (NCT01998841) evaluated crenezumab in asymptomatic PSEN1 E280A mutation carriers at high risk for autosomal-dominant AD (ADAD). Conducted in Colombia, 252 participants were enrolled, including 200 mutation carriers randomized to crenezumab or placebo for 260 weeks. The primary outcome is the change in cognitive performance, with secondary outcomes assessing disease progression, biomarkers, and safety. This trial aimed to determine if crenezumab can delay or prevent cognitive impairment in preclinical ADAD and enhance understanding of AD biomarkers. While the phase III trials indicated limited efficacy, ongoing studies may provide insights into its potential in autosomal-dominant AD. The results of the Colombian trial will be published publicly later (three clinical trials).

Gantenerumab (RO4909832) is an IgG1 monoclonal antibody binding to aggregated Aβ, facilitating clearance through Fc receptor-mediated phagocytosis [141,142]. Although it failed to meet primary endpoints in the DIAN-TU trial (NCT04623242) for patients with inherited AD, further studies have shown reductions in Aβ plaques and tau biomarkers, albeit without cognitive improvements. Moreover, gantenerumab was also evaluated in two phase III trials (GRADUATE I NCT03444870 and II NCT03443973) involving participants with early AD. The trials aimed to assess its ability to slow cognitive and functional decline. Although gantenerumab effectively reduced amyloid plaque burden and showed biomarker changes, such as lower cerebrospinal fluid phosphorylated tau and higher Aβ42 levels, it failed to demonstrate significant clinical benefits. At 116 weeks, the differences in cognitive decline, measured using the Clinical Dementia Rating Scale-Sum of Boxes (CDR-SB), were not statistically significant compared to placebo. Additionally, amyloid-related imaging abnormalities with edema (ARIA-E) were observed in 24.9% of participants, with symptomatic cases in 5.0%. Despite encouraging biomarker changes, gantenerumab’s lack of cognitive benefits highlights the need for further research to optimize its therapeutic potential (three clinical trials).

The advancement of passive immunotherapy, particularly through these mAbs, represents a significant step in addressing the amyloid hypothesis and the underlying pathology of AD. Despite the challenges associated with treatment efficacy and safety concerns, these therapies have opened new avenues for research and development in the quest to combat this debilitating disease. As ongoing clinical trials yield further insights, they will be pivotal in determining the optimal use of these therapies and their potential to alter the disease course for individuals with AD.

### 3.2. Limitations in Amyloid-Targeting Therapies

Recent failures in Aβ therapies highlight the need to explore alternative pathways for AD treatment. These pathways include tauopathy, neuroinflammation, mitochondrial dysfunction, and oxidative stress. The interconnected nature of these processes contributes to the progression of AD, with oxidative damage, inflammation, and metabolic dysregulation playing pivotal roles in neuronal degeneration. Mitochondrial dysfunction is a key factor in Alzheimer’s pathology, leading to the excessive production of reactive oxygen species and oxidative stress, which accelerates neuronal damage and exacerbates other pathological features such as amyloid β and tau accumulation [143,144]. Research suggests that targeting mitochondrial bioenergetics may provide a promising therapeutic approach by addressing oxidative stress and its downstream effects.

Oxidative stress and neuroinflammation are closely linked. Inflammatory responses in the brain drive increased ROS production, which in turn amplifies inflammation, creating a vicious cycle of damage. Microglial activation, a hallmark of AD, not only exacerbates oxidative stress but also promotes tau protein hyperphosphorylation, further driving disease progression. By targeting oxidative stress pathways, it may be possible to address both inflammation and tauopathy simultaneously [143].

Therapeutic strategies to combat oxidative stress include direct antioxidants, modulation of intracellular defense systems, nanotechnology-based interventions, immunomodulatory agents, and therapies targeting metal-protein interactions. Antioxidants such as vitamin E, curcumin, and α-lipoic acid aim to neutralize ROS and reduce oxidative damage. While some combinations, like vitamin E and vitamin C, have shown efficacy in reducing oxidative biomarkers in cerebrospinal fluid, others, including coenzyme Q10 and ω-3 fatty acids, have shown limited effects in clinical trials [145].

Enhancing endogenous antioxidant defenses has emerged as another promising approach. Molecules like MitoTEMPO and MitoQ target mitochondrial dysfunction, reducing oxidative damage and protecting neuronal health. Nanotechnology offers additional potential, with nanoparticles like cerium oxide mimicking endogenous antioxidant enzymes such as superoxide dismutase and catalase to detoxify ROS and protect neurons. Mesoporous silica nanoparticles are also being investigated for their ability to efficiently deliver therapeutic agents, thanks to their structural tunability [146,147,148].

Immunomodulatory agents, including fingolimod, tanshinone I, and ginsenoside Rg1, demonstrate anti-inflammatory effects and reduce oxidative stress by modulating microglial activation [149,150]. These compounds could provide protection against Alzheimer’s-related neurodegeneration by interrupting the damaging cycle of inflammation and oxidative damage. Similarly, metal protein attenuating compounds aim to disrupt abnormal metal–protein interactions that contribute to oxidative stress and cytotoxicity, showing promise in preventing lipid peroxidation and amyloid aggregation [151].

Animal studies have provided substantial evidence for the effectiveness of oxidative stress-targeting therapies in reducing lipid peroxidation, enhancing antioxidant defenses, and improving cognitive performance. Agents such as melatonin, imperatorin, and S-allyl cysteine have demonstrated these benefits in preclinical models. However, translating these findings to human trials has proven challenging, with some antioxidants showing inconsistent results or limited efficacy [152,153,154].

In addition to oxidative stress, tauopathy and microglia-mediated neuroinflammation play pivotal roles in AD progression. Tau hyperphosphorylation leads to neurofibrillary tangles, a hallmark of the disease, while microglial activation sustains a chronic inflammatory state, further exacerbating neuronal loss. Addressing these processes offers an opportunity to develop therapies targeting the core mechanisms of AD.

In the following, therapies designed to modulate tau pathology and microglial activity will be explored in greater detail, highlighting their potential to complement oxidative stress-targeting strategies and contribute to a multifaceted approach to AD treatment.

### 3.3. Immunotherapy Based on Targeting Tau

Immunotherapies targeting tau protein have emerged as a promising approach in the treatment of AD, given the pivotal role of neurofibrillary tangles composed of hyperphosphorylated tau (p-tau) in the pathogenesis of the disease. Tau is a cytoplasmic protein essential for stabilizing microtubules during neuronal development. In a healthy state, tau binds effectively to tubulin, promoting microtubule polymerization. However, in AD, hyperphosphorylation of tau leads to a loss of microtubule binding capability, resulting in the formation of neurofibrillary tangles and aggregates, which are correlated with cognitive decline in patients. This association underscores the therapeutic potential of strategies aimed at modulating tau pathology [155].

Three primary strategies are recognized in anti-tau therapies: preventing abnormal tau phosphorylation, inhibiting tau aggregation, and promoting the clearance of tau aggregates (Figure 4). The majority of current anti-tau agents in clinical development are immunotherapies, which can be classified into active and passive approaches.

#### 3.3.1. Active Immunotherapy

Two notable examples of active immunotherapy vaccines currently in development are AADvac1 and ACI-35, each designed to target specific tau epitopes associated with the pathological changes in AD.

AADvac1, developed by Axon Neuroscience, is a first-generation active immunotherapy that targets a specific 12-amino-acid sequence (KDNIKHVPGGGS) located within the microtubule-binding region of the tau protein. This vaccine has demonstrated a favorable safety profile, with rare adverse events reported among both immunized and placebo groups [156]. In the phase I trial (NCT02031198), AADvac1 treatment was associated with less brain atrophy and a reduction in cognitive decline among patients with mild-to-moderate AD. Notably, AADvac1 also significantly reduced levels of two cerebrospinal fluid (CSF) biomarkers indicative of AD pathology, p-tau181 and p-tau217. These encouraging findings prompted the transition of AADvac1 into phase II trials [157,158]. More recently, another phase II trial evaluating the long-term safety, tolerability, immunogenicity, and efficacy of AADvac1 in slowing cognitive decline was completed. While the results indicated that AADvac1 was safe and well tolerated, it did not demonstrate a significant improvement in cognitive function among a total of 196 patients [159]. Despite these mixed results, AADvac1 represents a significant step forward in active immunotherapy for AD, showcasing its potential in targeting tau pathology while emphasizing the necessity for further studies to confirm its efficacy.

ACI-35, developed by AC Immune, is an active immunotherapy vaccine designed to target hyperphosphorylated tau. The vaccine incorporates 16 copies of a synthetic tau peptide that specifically recognizes the pathological phosphorylation residues S396 and S404 of tau. A multicenter, double-blind, randomized phase I/II clinical trial (NCT04445831) was conducted in Finland to assess the safety and efficacy of ACI-35 in patients with mild-to-moderate AD. The trial was completed on 16 July 2024, but the results are not yet publicly available. The eventual findings from this trial are expected to provide critical insights into the safety, immunogenicity, and potential efficacy of ACI-35 in targeting hyperphosphorylated tau and slowing disease progression in AD.

As research continues into both AADvac1 and ACI-35, these active immunotherapy vaccines represent an evolving landscape in the fight against AD, reflecting the potential to harness the immune system to combat tau pathology and slow cognitive decline.

#### 3.3.2. Passive Immunotherapy

Semorinemab

Semorinemab (RO705705) is a humanized monoclonal antibody characterized by its IgG4 isotype backbone, specifically targeting extracellular tau. It binds to all six human tau isoforms and offers neuroprotective effects [160]. Preclinical studies demonstrated a significant reduction in tau accumulation in mouse models, suggesting potential efficacy. However, clinical trials have not shown effectiveness signals for AD. The phase II Lauriet study evaluated semorinemab in patients with mild-to-moderate AD. Participants received 4500 mg of semorinemab or placebo every 4 weeks for up to 60 weeks, with coprimary endpoints assessing changes in cognition (ADAS-Cog11) and daily functioning (ADCS-ADL). Semorinemab showed a significant 42.2% reduction in cognitive decline on the ADAS-Cog11 compared to placebo (*p* = 0.0008) but had no effect on daily functioning (ADCS-ADL) or secondary endpoints (MMSE, CDR-SB). Semorinemab was safe and well tolerated. While the study met one cognitive endpoint, the lack of functional improvement highlights the need for further exploration of anti-tau therapies in AD [161]. Moreover, a report in *Nature Reviews Drug Discovery* indicated that semorinemab did not alleviate AD symptoms and highlighted the failure of a prior phase I trial. The failure of the semorinemab trial can be attributed to the lack of significant improvements in cognitive or functional outcomes in patients with prodromal to mild AD, despite reductions in tau biomarkers. The trial’s design, including its endpoints and patient population, may have contributed to this lack of efficacy. The timing of intervention (early-stage disease) and the complexity of tau’s role in Alzheimer’s pathology also likely influenced the results. (NCT02754830) [162].

BIIB076 and Gosuranemab

Both BIIB076 (NI-105) and gosuranemab (BIIB092) have been developed by Biogen.

BIIB076 is a human recombinant IgG1 monoclonal antibody targets the mid-domain of tau (amino acids 125–131). Preclinical studies indicated its ability to block tau aggregation. A phase I trial (NCT03056729) has been completed, but development was halted for undisclosed reasons (one clinical trial).

Gosuranemab recognizes the extracellular N-terminal tau. It significantly reduced unbound N-terminal tau fragments in cerebrospinal fluid (CSF) during a phase II trial for progressive supranuclear palsy (NCT03068468), showing a 98% reduction compared to an 11% increase in the placebo group. However, this neutralization did not translate into clinical effectiveness, leading to the discontinuation of its development in early AD participants due to lack of efficacy. The failure of the gosuranemab (BIIB092) trial can be attributed to several factors. Despite demonstrating target engagement by lowering N-terminal tau fragments in cerebrospinal fluid (CSF), gosuranemab did not show a significant effect on tau accumulation in the brain or on cognitive decline. One key issue may have been the choice of the N-terminus of tau as the target, as other anti-tau antibodies have also failed, suggesting that this may not be the most effective approach. Additionally, the timing of intervention in patients with mild cognitive impairment or mild AD may have limited the therapeutic impact, as tau pathology may be too advanced for intervention at this stage. Furthermore, the clinical endpoints used in the study, such as the Clinical Dementia Rating-Sum of Boxes (CDR-SB), may not have been sensitive enough to detect subtle changes in disease progression, contributing to the lack of observed efficacy. These challenges highlight the complexities of targeting tau in AD and the need for further exploration of optimal treatment strategies [163,164]. In another study by Kim and colleagues, it was suggested that gosuranemab treatment may trigger glial responses, including tau accumulation within astrocytic lysosome [165].

Tilavonemab

Tilavonemab (ABBV-8E12) targets the aggregated extracellular form of pathological tau, specifically binding to the N-terminus. Developed by C2N Diagnostics and AbbVie, it has shown safety in a phase I trial (NCT02880956) [166]. However, the subsequent phase II trial assessing efficacy and safety in 453 patients with early AD did not yield the expected outcomes, resulting in discontinuation (NCT02880956). Another phase II trial evaluating long-term safety and tolerability discontinued because of lack of efficacy in the parent study (NCT03712787). These trials underscore the challenges in developing effective passive immunotherapies targeting tau, as well as the need for novel approaches or improved study designs.

Bepranemab

Bepranemab (UCB0107) is a humanized IgG4 monoclonal antibody targeting the mid-region of tau (amino acids 235–250). It may be more effective in interrupting the cell-to-cell transmission of pathogenic tau than antibodies targeting the N-terminus [167]. A phase II trial currently enrolling participants (NCT04867616) aims to evaluate its efficacy, safety, and tolerability in patients with mild AD, with completion expected in November 2025. Ongoing studies will help determine bepranemab’s potential to slow disease progression by mitigating tau propagation.

Zagotenemab

Zagotenemab (LY3303560), derived from the mouse monoclonal antibody MCI-1, targets residues 7–9 and 312–322 of tau. After promising preclinical results, a phase II trial (NCT03518073) in early symptomatic AD patients failed to meet its primary endpoint, resulting in the termination of zagotenemab’s development. This highlights the difficulties in translating preclinical successes into clinical benefits for AD patients.

JNJ-63733657

This humanized IgG1 monoclonal antibody targets the microtubule-binding region of tau with high affinity for pThr217. It has shown dose-dependent reductions in phosphorylated tau (pTau) in CSF and is currently undergoing phase II trials to evaluate its effects on cognitive decline in AD patients (NCT04619420). Preliminary biomarker results are promising, but clinical efficacy in slowing cognitive decline remains to be demonstrated.

E2814

E2814 is an IgG1 monoclonal antibody targeting the HVPGG epitope in the mid-domain of tau [168]. A current phase I/II trial (NCT04971733) showed that patients treated with E2814 experienced approximately 75% and 50% reductions in CSF MTBR-tau243 and p-tau217, respectively. Tau PET imaging also indicated stabilized or decreased brain tau accumulation in DIAD participants. These findings suggest that E2814 inhibits tau propagation and accumulation (clinical trial). Further investigations are underway in the phase II/III Tau NexGen study (NCT05269394) for DIAD and the phase II 202 study (NCT06602258) for sporadic early AD. These early results indicate significant promise in mitigating tau pathology, potentially altering disease progression.

Lu AF87908

This humanized IgG1 antibody specifically targets tau phosphorylated at residues 396 and 404. A phase I trial (NCT04149860) assessed its safety in healthy participants and AD patients, with early results suggesting it may effectively reduce phosphorylated tau levels but the final results are not yet publicly available. Lu AF87908 remains in the early stages of development, with further data needed to evaluate its safety and efficacy.

PNT001

PNT001 is a human IgG4 monoclonal antibody that targets the cis-isomer of tau phosphorylated at T231 (cis-pT231 tau). It has undergone a phase I clinical trial (NCT04096287) to evaluate its safety, tolerability, pharmacokinetics, and immunogenicity in healthy adults.

In this randomized, double-blind, placebo-controlled study, 50 healthy volunteers were enrolled (49 dosed: 36 with PNT001 and 13 with placebo). Single intravenous doses ranging from 33 to 4000 mg were well tolerated. Only three mild, resolved adverse events occurred at lower doses, with no treatment-related serious adverse events or dose-limiting toxicities reported.

Pharmacokinetic analysis demonstrated dose-proportional serum levels, with a half-life ranging from 23.8 to 33.8 days. Cerebrospinal fluid (CSF) concentrations at higher doses (900–4000 mg) were sufficient for target engagement. One anti-drug antibody response was observed. These findings support the progression of PNT001 to further clinical trials in patients with tauopathies.

RG7345

RG7345 is an anti-tau monoclonal antibody that recognizes tau phosphorylated at the S422 site. Its phase I trial (NCT02281786) assessing safety in healthy participants was completed in 2015, but relevant data have not been released to the public. The lack of publicly available data limits insights into the antibody’s potential efficacy or safety profile in addressing tau pathology in AD.

### 3.4. Immunotherapy Targeting Microglia

Immunotherapy strategies focusing on microglia are emerging as promising avenues for addressing AD [169,170]. In the context of AD, microglia recognize Aβ and phosphorylated tau proteins as damage-associated molecular patterns through receptors like TLR-4 [171]. This recognition triggers the release of inflammatory factors, which can exacerbate Aβ deposition and tau pathology, creating a vicious cycle that accelerates disease progression [172].

Recent studies have elucidated the interplay between microglia and these pathological proteins. A PET imaging study involving 130 individuals highlighted that the interaction between Aβ and activated microglia significantly influences the spread of tau pathology across Braak stages, indicating a close relationship between microglial activation and AD progression [170]. Genome-wide association studies (GWAS) have identified several AD risk genes, including TREM2, which are highly expressed in microglia. These findings suggest that modulating microglial function through targeted therapies could be a viable strategy for treating AD [169].

One promising candidate in this area is AL002, a humanized monoclonal IgG1 antibody that targets TREM2, a receptor selectively expressed by microglia [173]. Certain TREM2 variants have been linked to an increased risk of late-onset AD [174].

AL002 is designed to activate TREM2 signaling, leading to microglial proliferation and a subsequent reduction in AD pathology in animal models. Preclinical studies demonstrated robust target engagement and microglial activation, which supported its advancement to clinical trials. Encouraging preclinical results have led to its evaluation in a phase II clinical trial, aiming to assess efficacy and safety in 265 participants with early-stage AD. The INVOKE-2 phase II clinical trial evaluated the safety and efficacy of AL002 in individuals with early AD. While treatment with AL002 demonstrated sustained target engagement and microglial activation, it did not meet the primary endpoint of slowing clinical progression, as measured using the Clinical Dementia Rating Sum of Boxes (CDR-SB). No treatment effects were observed on secondary clinical and functional endpoints as well as on Alzheimer’s fluid biomarkers or amyloid PET imaging. Additionally, MRI findings revealed amyloid-related imaging abnormalities (ARIA) and infusion-related reactions, with ARIA predominantly occurring in participants receiving AL002. Given the lack of clinical efficacy and biomarker improvements, the results suggest that AL002 does not effectively slow AD progression (NCT05744401, NCT04592874 and NCT03635047).

Another interesting approach involves daratumumab, a human antibody targeting CD38 that has received FDA approval for the treatment of multiple myeloma. Notably, daratumumab can cross the blood–brain barrier and has immunomodulatory effects on non-plasma cells expressing CD38. Research has shown that CD38+ CD8+ T-cells are elevated in early AD patients, suggesting these cells could migrate to the central nervous system and contribute to neurotoxicity [175,176].

A small, open-label pilot study evaluated daratumumab in patients with mild to moderate AD to assess target engagement, safety, and potential efficacy. Daratumumab (SC 1800 mg) was administered weekly for 8 weeks, then biweekly for 16 weeks. Flow cytometry revealed a significant reduction in CD38+ CD8+ CD4− T cells after 24 weeks, with the effect persisting for 11 weeks post-treatment. The treatment was well tolerated, with no hematological toxicity or unexpected adverse events. However, responder analysis showed no improvement in cognitive outcome measures, indicating a lack of efficacy in slowing cognitive decline (NCT04070378).

Sodium oligomannate, a compound derived from marine brown algae, represents another noteworthy development [177]. This compound influences gut microbiota and its associated effects on microglial activation and inflammation. While not a traditional immunotherapy, sodium oligomannate has been shown to reduce microglial activation and neuroinflammation, thereby slowing cognitive decline [178]. The drug received conditional approval in China (NCT05908695) for the treatment of mild to moderate AD in November 2019 and was included in the national medical insurance catalogue in December 2021 [177]. An international phase III trial is currently exploring the efficacy of sodium oligomannate in mild to moderate AD patients across North America, Europe, and Asia [173]. While the global trial is ongoing, initial findings from the Chinese population suggest a modest benefit in slowing cognitive decline, warranting further investigation.

Another therapeutic candidate developed by Denali Therapeutics, DNL919, incorporates a transferrin-receptor binding sequence to enhance transcytosis across the blood–brain barrier (BBB). A phase I trial was performed for this candidate as a randomized, double-blind, placebo-controlled, single ascending dose (SAD) study aimed at evaluating its safety, tolerability, pharmacokinetics (PK), and pharmacodynamics (PD) in healthy participants. Conducted in the Netherlands, the study enrolled 47 participants aged 18 to 55 years with a body mass index (BMI) between 18.5 and less than 30 kg/m^2^. Both male and female participants (the latter of non-childbearing potential) were eligible for the trial. The trial excluded individuals with a history of significant neurological, psychiatric, cardiovascular, or other major health disorders. It began in July 2022, reached primary completion in June 2023, and concluded in the same month. Although the trial’s results have not yet been made publicly available, its completion represents a key milestone in the clinical evaluation of BBB-crossing immunotherapies for AD (NCT05744401) [179].

Ultimately, targeting neuroinflammation through passive immunotherapy represents a novel and promising strategy for modifying the course of AD [174]. By enhancing microglial function and reducing neuroinflammation, these therapies aim to interrupt the pathological cycles associated with Aβ and tau, ultimately leading to improved outcomes for patients with AD. As these therapies advance through clinical trials, they could pave the way for new treatment options that address the underlying neuroinflammatory processes in neurodegenerative diseases. Ongoing clinical trials will provide crucial insights into the efficacy and safety of these innovative treatments, potentially leading to new therapeutic strategies against AD.

## 4. Challenges in Immunotherapy of AD

The pursuit of effective immunotherapy for AD encounters a myriad of challenges that significantly impede the translation of preclinical promise into clinical success. Despite considerable advancements in understanding AD pathology, several complexities related to the disease and the intricacies of immune responses create substantial hurdles for developing successful immunotherapeutic strategies. Herein, the primary challenges faced in this area will be analyzed.

### 4.1. Limitations of Animal Models

Although numerous animal models are available for AD research, none can fully replicate the complex neuropathological and behavioral features seen in humans, such as the absence of Aβ plaques in tau-based models or the lack of neurofibrillary tangles (NFTs) in APP-based models. This mismatch between animal and human disease raises concerns about the relevance of preclinical findings and often leads to high failure rates in clinical trials. Moreover, some potential drugs have been tested in clinical trials without robust preclinical data, increasing the likelihood of failure. To improve translational success, therapeutic strategies should be tested across various AD models, and standardized protocols should be followed to ensure the consistency and reproducibility of results. Sex-differences, age-dependent disease progression, and interspecies variability must also be considered when designing studies. Despite these limitations, animal models remain valuable tools for advancing our understanding of AD and developing new treatments [180].

### 4.2. Suboptimal Immune Responses

Many investigational immunotherapies target specific pathological proteins associated with AD, yet the ability of these therapies to elicit sufficiently robust immune responses remains uncertain. A key challenge lies in understanding the pharmacokinetics and pharmacodynamics of these therapies, particularly how they stimulate humoral immune responses in humans. There is a pressing need for detailed studies that investigate optimal dosing regimens, timing of administration, and the characteristics of antibodies produced. Such insights are crucial for maximizing therapeutic effectiveness and ensuring meaningful clinical outcomes [181].

### 4.3. Complexities in Preventive Clinical Trials

Preventive clinical trials for AD are inherently complex, requiring significant time and resources. Enrolling asymptomatic individuals at risk of developing the disease presents unique challenges, including the need for costly and invasive diagnostic procedures, such as brain imaging and cerebrospinal fluid analysis. These logistical burdens can lead to delays in trial progress and increased financial strain, ultimately hindering the advancement of effective immunotherapies. Moreover, identifying the right patient population for such trials complicates study design and execution [182].

### 4.4. Insufficient Characterization of Immune Responses

Current knowledge regarding the immune responses elicited by active immunotherapies remains limited. While ongoing clinical trials have demonstrated acceptable safety profiles for several therapies, the intricacies of the immune response particularly the quality and functionality of the antibodies generated are not yet fully understood. Most studies tend to focus on antibody specificity while neglecting the broader immune context. A comprehensive understanding of the immune mechanisms at play is essential for refining treatment strategies and enhancing immunotherapeutic efficacy [183].

### 4.5. Identification of Therapeutic Targets

A critical step in developing effective disease-modifying interventions is the accurate identification of therapeutic targets. The roles of Aβ and tau proteins in AD progression have long been debated, with postmortem studies indicating that not all patients with Aβ deposits exhibit cognitive impairment. This inconsistency necessitates a revaluation of Aβ and tau as optimal targets. In addition, the exploration of alternative molecules that can contribute to AD progression is relevant. Additionally, the distinction between the various forms of Aβ, e.g., toxic oligomers versus less harmful aggregates, is vital for guiding targeted immunotherapy development [122].

### 4.6. Challenges in Delivery Drugs Across the Blood–Brain Barrier

A significant obstacle to effective immunotherapy is the limited ability of therapeutic agents to penetrate the blood–brain barrier (BBB). Most vaccines and antibodies administered peripherally achieve minimal central nervous system penetration. Innovations in bioengineering and drug delivery systems, such as biomimetic nanoparticles, hold promise for enhancing the delivery efficiency of these agents across the BBB, ensuring that adequate drug concentrations reach the brain [184].

### 4.7. Mitigating Adverse Effects

Developing safe immunotherapy options is paramount, particularly concerning vaccines. Care must be taken to avoid autoimmune T-cell activation and to minimize Fc-mediated inflammatory responses using human antibodies. Strategies that focus on specific B cell epitopes, as well as exploring advanced antibody formats like single-chain antibodies and Fc-deglycosylated antibodies, may help to reduce adverse reactions and improve the safety profile of immunotherapies [185].

### 4.8. Need for Comprehensive Treatment Approaches

Given the multifaceted nature of AD, relying on therapies that target only one aspect of the disease may not yield optimal clinical outcomes. A combinatorial approach that addresses both Aβ and tau pathologies simultaneously may offer a more effective strategy. However, careful consideration must be given to the potential increase in T-helper responses and antibody concentrations that can arise from such combination therapies, necessitating meticulous management to optimize both safety and efficacy [186].

## 5. Conclusions

Overcoming the challenges associated with immunotherapy for AD is pivotal in advancing treatment strategies and improving patient outcomes. Despite significant progress in understanding the immune mechanisms involved in AD, the translation of immunotherapeutic approaches from preclinical models to clinical applications has been met with various hurdles. The predictive validity of these preclinical models remains a critical limitation. Many animal models fail to fully replicate the complexity of human AD pathology, particularly the late-onset, sporadic nature of the disease. By enhancing the fidelity of these models, researchers can improve the translation of experimental results into clinical benefits.

A deeper understanding of the immune responses underlying AD progression, particularly the role of microglia, peripheral immune cells, and the gut–brain axis, is also essential. Current immunotherapies have primarily focused on Aβ and tau pathology, but emerging evidence suggests that neuroinflammation plays a central role in driving disease progression. Understanding how to modulate immune responses without triggering adverse effects, such as excessive neuroinflammation or immune suppression, is key to developing safer and more effective treatments. Additionally, research into patient stratification and biomarker development will allow for more personalized approaches, tailoring immunotherapies to the specific needs and disease stages of individual patients.

Innovative drug delivery mechanisms will play an increasingly important role in the success of future immunotherapies. Many current therapies face challenges in delivering therapeutic agents across the blood–brain barrier, reducing their efficacy. Novel strategies, such as nanotechnology-based delivery systems and direct intranasal administration, could help overcome these obstacles, ensuring that treatments reach their intended targets in the brain more effectively. In parallel, combination therapies that address multiple aspects of AD pathology, such as Aβ, tau, and neuroinflammation, hold promise for achieving greater clinical efficacy. These multifaceted approaches could potentially slow or even halt disease progression by addressing the various overlapping mechanisms involved in AD. The development of these advanced immunotherapeutic strategies offers hope for halting the progression of AD, though reversing the disease remains unachieved, and also holds broader implications for the treatment of other neurodegenerative diseases. Ongoing breakthroughs in immune modulation, novel drug delivery systems, and personalized treatment strategies are driving the field toward new horizons, offering tangible hope for improved patient outcomes. The increasing focus on the immune system’s role in neurodegeneration is revealing promising therapeutic targets, making disease-modifying treatments for AD a more achievable goal.

In conclusion, the path forward for immunotherapy in AD involves a combination of improved preclinical models, a refined understanding of immune mechanisms, innovative delivery technologies, and personalized, multi-targeted treatment approaches. As these elements come together, the potential to transform the landscape of AD treatment is clearer than ever, with the promise of not only slowing disease progression but also fundamentally changing the way we approach neurodegenerative diseases. If these challenges are addressed, immunotherapy has the potential to offer new hope for patients and their families. Ultimately, these advancements will be instrumental in moving from symptom management to disease modification, bringing us closer to the goal of preventing and curing AD.

## Figures and Tables

**Figure 1 cells-14-00264-f001:**
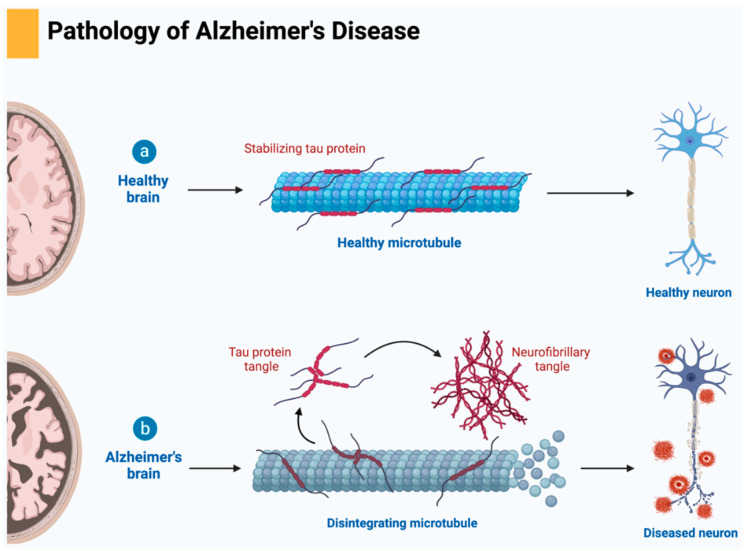
Tau and neurofibrillary tangle pathology in AD. The figure illustrates the role of tau protein and neurofibrillary tangles in the pathology of AD. In AD, tau protein undergoes critical modifications, including hyperphosphorylation, which causes it to detach from microtubules and aggregate into toxic neurofibrillary tangles. These tangles impair neuronal function and contribute to synaptic dysfunction. The comparison is made between the pathological features of an AD brain and a healthy brain, highlighting the effects of tau pathology at various magnification levels.

**Figure 2 cells-14-00264-f002:**
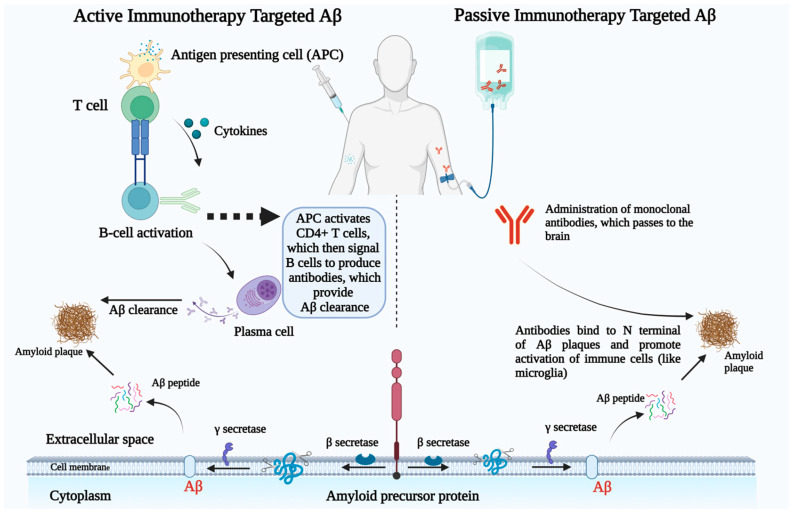
Active and passive immunotherapies as potential strategies to treat AD by targeting Aβ plaques. Active immunotherapy (**left**) entails vaccinating the patient with Aβ fragments to stimulate their immune system. Antigen-presenting cells (APCs) present the Aβ antigen to CD4+ T cells, which subsequently activate B cells. These activated B cells mature into plasma cells that produce antibodies specifically targeting Aβ. These antibodies bind to the plaques and promote their clearance through various mechanisms. On the other hand, passive immunotherapy (**right**) involves administering mAbs capable of crossing the blood–brain barrier (BBB) and directly binding to Aβ plaques. This interaction activates microglia, which then carry out phagocytosis to clear the plaques from the brain.

**Figure 3 cells-14-00264-f003:**
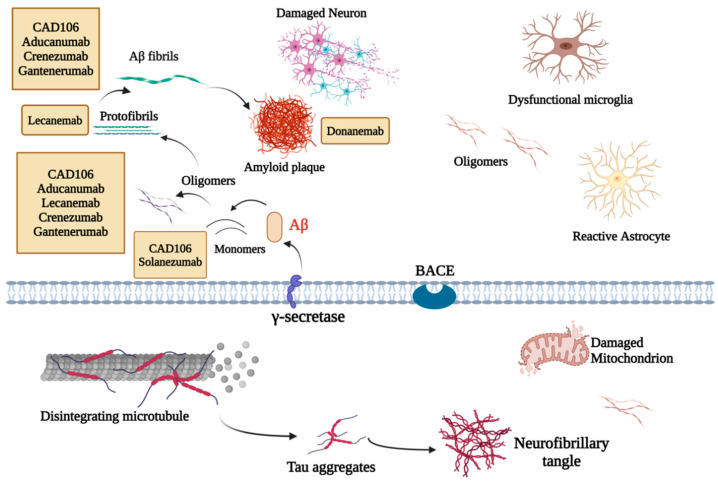
Immunotherapy approaches targeting Aβ in AD. This figure illustrates various immunotherapeutic strategies aimed at different forms of Aβ aggregates, including monomers, oligomers, protofibrils, fibrils, and plaques. These therapies are designed to prevent the formation of Aβ aggregates, promote their clearance, or disrupt their toxic effects. Notably, gantenerumab and aducanumab have been reported to target plaques, fibrils, and oligomers. Additionally, the figure depicts key pathological features associated with AD progression. Damaged mitochondria are shown as contributing to neuronal dysfunction, driven by oxidative stress and energy deficits. Disintegrating microtubules, depicted as fragmented structures, result from tau hyperphosphorylation, which disrupts axonal transport. This process contributes to the formation of neurofibrillary tangles, insoluble aggregates of hyperphosphorylated tau that further impair neuronal function. The figure highlights the complex interplay among Aβ aggregation, tau pathology, and cellular dysfunction, emphasizing their collective role in AD progression.

**Figure 4 cells-14-00264-f004:**
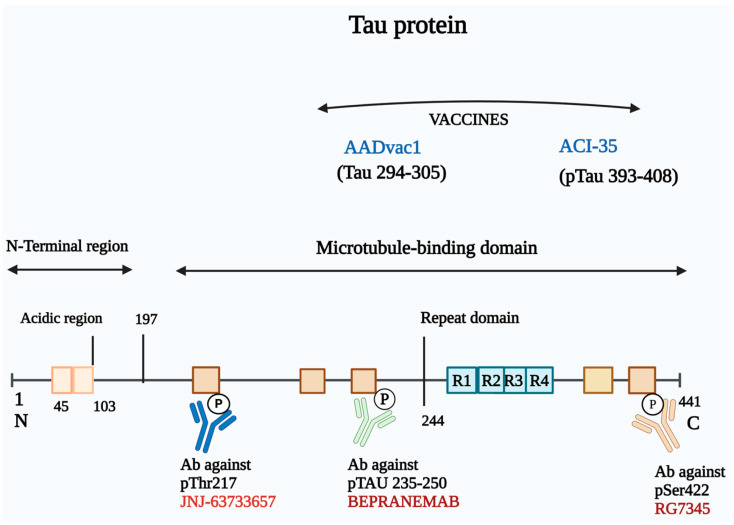
Tau-targeted immunotherapy strategies. Anti-Tau antibodies produced through vaccines (active immunotherapy) and or human anti-Tau antibodies (passive immunotherapy) can target specific regions, sites, or phosphorylated (P) Tau to influence its aggregation. Some of the drugs used are shown in the figure.

## Data Availability

Not applicable.

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
