# Peer review of "Immune Modulation in Alzheimer’s Disease: From Pathogenesis to Immunotherapy"

_cells, 2025, doi:10.3390/cells14040264_

Round 1

Reviewer 1 Report

Comments and Suggestions for Authors

The article emphasizes the amyloid cascade hypothesis as the primary driver of Alzheimer's disease (AD) pathogenesis. This is an interesting and timing review on AD immunotherapy. However, there are some major concerns need to be addressed by the authors:

1, the hypothesis has guided AD research for decades, However, recent failures in amyloid-targeting therapies (e.g., solanezumab and crenezumab) suggest that alternative pathways, such as tauopathy, inflammation, and metabolic dysfunction, might play equally crucial roles. A more balanced discussion incorporating these perspectives could enhance the article's depth.

2, The review lists numerous immunotherapy trials (e.g., aducanumab, donanemab) but could benefit from a more critical evaluation of their controversial results. For instance, aducanumab's approval by the FDA faced significant backlash due to mixed clinical outcomes and safety concerns. The article could delve deeper into the discrepancies between the ENGAGE and EMERGE trials to highlight the complexities and uncertainties in immunotherapy for AD.

3, While the article briefly touches on the role of peripheral immune cells (e.g., monocytes, neutrophils), it does not sufficiently explore their interactions with the central nervous system (CNS). Recent studies suggest that systemic inflammation and gut microbiota can influence AD progression through the microbiota-gut-brain axis. The authors should expand discussion on microbiota-gut-brain axis, which could provide a more comprehensive view of immune modulation.

4, The article mentions APOE ε4 as a risk factor but does not extensively discuss the implications of genetic and biomarker heterogeneity in tailoring immunotherapy. A deeper exploration of personalized approaches, such as stratifying patients based on genetic markers or cerebrospinal fluid (CSF) biomarkers, could enhance the relevance of immunotherapies.

5, The section on gut microbiota's role in AD is intriguing but speculative. While emerging data suggest that gut dysbiosis may contribute to neuroinflammation, the article lacks sufficient clinical evidence to support these claims robustly. The authors should include more references to longitudinal human studies or meta-analyses would strengthen this section.

6, The discussion on tau-targeted therapies, such as semorinemab and gosuranemab, mentions trial failures without dissecting the reasons behind them. Addressing whether these failures stem from the antibody design, patient selection, or trial endpoints would provide valuable insights into the challenges of targeting tau.

Author Response

Reviewer 1

Comment #1

The hypothesis has guided AD research for decades, However, recent failures in amyloid-targeting therapies (e.g., solanezumab and crenezumab) suggest that alternative pathways, such as tauopathy, inflammation, and metabolic dysfunction, might play equally crucial roles. A more balanced discussion incorporating these perspectives could enhance the article's depth.

Dear Reviewer, Thank you for your valuable feedback. We agree that recent failures in amyloid-targeting therapies highlight the importance of exploring alternative pathways in AD. In response, we have expanded the section on limitations to include tauopathy, neuroinflammation, and mitochondrial dysfunction, emphasizing their interconnected roles in AD progression. We have also added a more balanced discussion on oxidative stress and its link to inflammation and tau hyperphosphorylation, suggesting that targeting oxidative stress may address both inflammation and tauopathy simultaneously. Additionally, we have elaborated on therapeutic strategies involving antioxidants, immunomodulatory agents, and nanotechnology, which show promise in combating these mechanisms. We hope these revisions enhance the depth of the article by considering a broader range of therapeutic targets in AD treatment.

3.2.Limitations in amyloid-targeting therapies

Recent failures in amyloid-targeting therapies highlight the need to explore alternative pathways for Alzheimer's disease treatment. These pathways include tauopathy, neuroinflammation, mitochondrial dysfunction, and oxidative stress. The interconnected nature of these processes contributes to the progression of AD, with oxidative damage, inflammation, and metabolic dysregulation playing pivotal roles in neuronal degeneration. Mitochondrial dysfunction is a key factor in Alzheimer’s pathology, leading to the excessive production of reactive oxygen species and oxidative stress, which accelerates neuronal damage and exacerbates other pathological features such as amyloid-beta and tau accumulation (Dash et al, 2024) (Rummel et al, 2022). Research suggests that targeting mitochondrial bioenergetics may provide a promising therapeutic approach by addressing oxidative stress and its downstream effects.

Dash UC, Bhol NK, Swain SK, Samal RR, Nayak PK, Raina V, Panda SK, Kerry RG, Duttaroy AK, Jena AB. Oxidative stress and inflammation in the pathogenesis of neurological disorders: Mechanisms and implications. Acta Pharm Sin B. 2024; https://doi.org/10.1016/j.apsb.2024.10.004

Rummel NG, Butterfield DA. Altered metabolism in Alzheimer disease brain: Role of oxidative stress. Antioxid Redox Signal. 2022 Jun 3;36(16-18):1289–1305. doi: 10.1089/ars.2021.0177.

Oxidative stress and neuroinflammation are closely linked. Inflammatory responses in the brain drive increased ROS production, which in turn amplifies inflammation, creating a vicious cycle of damage. Microglial activation, a hallmark of AD, not only exacerbates oxidative stress but also promotes tau protein hyperphosphorylation, further driving disease progression. By targeting oxidative stress pathways, it may be possible to address both inflammation and tauopathy simultaneously (Dash et al, 2024).

Therapeutic strategies to combat oxidative stress include direct antioxidants, modulation of intracellular defense systems, nanotechnology-based interventions, immunomodulatory agents, and therapies targeting metal-protein interactions. Antioxidants such as vitamin E, curcumin, and α-lipoic acid aim to neutralize ROS and reduce oxidative damage. While some combinations, like vitamin E and vitamin C, have shown efficacy in reducing oxidative biomarkers in cerebrospinal fluid, others, including coenzyme Q10 and ω-3 fatty acids, have shown limited effects in clinical trials (Aborode et al, 2022).

Aborode AT, Pustake M, Awuah WA, Alwerdani M, Shah P, Yarlagadda R, Ahmad S, Silva Correia IF, Chandra A, Nansubuga EP, Abdul-Rahman T, Mehta A, Ali O, Amaka SO, Zuñiga YMH, Shkodina AD, Inya OC, Shen B, Alexiou A. Targeting oxidative stress mechanisms to treat Alzheimer's and Parkinson's disease: A critical review. Oxid Med Cell Longev. 2022 Jul 31;2022:7934442. doi: 10.1155/2022/7934442.

Enhancing endogenous antioxidant defenses has emerged as another promising approach. Molecules like MitoTEMPO and MitoQ target mitochondrial dysfunction, reducing oxidative damage and protecting neuronal health. Nanotechnology offers additional potential, with nanoparticles like cerium oxide mimicking endogenous antioxidant enzymes such as superoxide dismutase and catalase to detoxify ROS and protect neurons. Mesoporous silica nanoparticles are also being investigated for their ability to efficiently deliver therapeutic agents, thanks to their structural tunability (McManus et al, 2011)(Reddy et al, 2012)(Augustyniak et al, 2010).

McManus MJ, Murphy MP, Franklin JL. The mitochondria-targeted antioxidant MitoQ prevents loss of spatial memory retention and early neuropathology in a transgenic mouse model of Alzheimer's disease. J Neurosci. 2011 Nov 2;31(44):15703-15. doi: 10.1523/JNEUROSCI.0552-11.2011.

Reddy P. H., Tripathi R., Troung Q., et al. Abnormal mitochondrial dynamics and synaptic degeneration as early events in Alzheimer’s disease: implications to mitochondria-targeted antioxidant therapeutics. Biochimica et Biophysica Acta (BBA)-Molecular Basis of Disease . 2012;1822(5):639–649. doi: 10.1016/J.BBADIS.2011.10.011.

Augustyniak A., Bartosz G., Čipak A., et al. Natural and synthetic antioxidants: an updated overview. Free Radical Research . 2010;44(10):1216–1262. doi: 10.3109/10715762.2010.508495.

Immunomodulatory agents, including Fingolimod, Tanshinone I, and Ginsenoside Rg1, demonstrate anti-inflammatory effects and reduce oxidative stress by modulating microglial activation (B. Martinez et al, 2018)(Rekatsina et al, 2019). These compounds could provide protection against Alzheimer’s-related neurodegeneration by interrupting the damaging cycle of inflammation and oxidative damage. Similarly, metal-protein attenuating compounds aim to disrupt abnormal metal-protein interactions that contribute to oxidative stress and cytotoxicity, showing promise in preventing lipid peroxidation and amyloid aggregation (V. B. Kenche et l, 2011).

  1. Martinez and P. Peplow, “Neuroprotection by immunomodulatory agents in animal models of Parkinson’s disease,”Neural Regeneration Research, vol. 13, no. 9, pp. 1493–1506,2018.

Rekatsina M, Paladini A, Piroli A, Zis P, Pergolizzi JV, Varrassi G. Pathophysiology and therapeutic perspectives of oxidative stress and neurodegenerative diseases: A narrative review. Adv Ther. 2019 Nov 28;37(1):113–139. doi: 10.1007/s12325-019-01148-5.

  1. B. Kenche and K. J. Barnham, “Alzheimer’s disease &metals: therapeutic opportunities,” British Journal of Pharmacology,vol. 163, no. 2, pp. 211–219, 2011.

Animal studies have provided substantial evidence for the effectiveness of oxidative stress-targeting therapies in reducing lipid peroxidation, enhancing antioxidant defenses, and improving cognitive performance. Agents such as melatonin, imperatorin, and S-allyl cysteine have demonstrated these benefits in preclinical models. However, translating these findings to human trials has proven challenging, with some antioxidants showing inconsistent results or limited efficacy (Z. Feng et al, 2006) (B. Budzynska et al, 2015)(H. Javed et al, 2011).

  1. Feng, C. Qin, Y. Chang, and J. T. Zhang, “Early melatonin supplementation alleviates oxidative stress in a transgenic mouse model of Alzheimer’s disease,” Free Radical Biology & Medicine, vol. 40, no. 1, pp. 101–109, 2006.
  2. Budzynska, A. Boguszewska-Czubara, M. Kruk-Slomka et al., “Effects of imperatorin on scopolamine-induced cognitive impairment and oxidative stress in mice,” Psychopharmacology, vol. 232, no. 5, pp. 931–942, 2015.
  3. Javed, M. M. Khan, A. Khan et al., “S-allyl cysteine attenuates oxidative stress associated cognitive impairment and neurodegeneration in mouse model of streptozotocin-induced experimental dementia of Alzheimer’s type,” Brain Research,vol. 1389, pp. 133–142, 2011.

In addition to oxidative stress, tauopathy and microglia-mediated neuroinflammation play pivotal roles in AD progression. Tau hyperphosphorylation leads to neurofibrillary tangles, a hallmark of the disease, while microglial activation sustains a chronic inflammatory state, further exacerbating neuronal loss. Addressing these processes offers an opportunity to develop therapies targeting the core mechanisms of AD.

In the following, therapies designed to modulate tau pathology and microglial activity will be explored in greater detail, highlighting their potential to complement oxidative stress-targeting strategies and contribute to a multifaceted approach to AD treatment.

Reviewer 1

Comment #2

2) The review lists numerous immunotherapy trials (e.g., aducanumab, donanemab) but could benefit from a more critical evaluation of their controversial results. For instance, aducanumab's approval by the FDA faced significant backlash due to mixed clinical outcomes and safety concerns. The article could delve deeper into the discrepancies between the ENGAGE and EMERGE trials to highlight the complexities and uncertainties in immunotherapy for AD.

We thank the reviewer for raising an important point about the controversial results of immunotherapy trials, specifically aducanumab. As mentioned, the discrepancies between the ENGAGE and EMERGE trials are crucial for understanding the complexities and uncertainties in AD immunotherapy. While EMERGE demonstrated a modest reduction in clinical decline, ENGAGE failed to show any clinical benefit, raising concerns about the reliability and replicability of these findings.

Biogen's decision to pursue regulatory approval using post hoc analyses, which deviated from the prespecified statistical plan, has indeed been contentious. The modest effect size in EMERGE, combined with inconsistent results between trials, and the reliance on amyloid plaque reduction as a surrogate endpoint, have sparked criticism regarding the robustness of the evidence supporting aducanumab’s approval.

We agree with the reviewer that a deeper evaluation of these issues is necessary to understand the limitations of amyloid-targeting therapies. This underscores the need for more rigorous trial designs, adherence to prespecified analysis plans, and greater transparency in the regulatory process to ensure the safety and efficacy of treatments for AD.

Although a phase II study was bypassed due to promising phase I results, two identically designed phase III trials, ENGAGE (NCT02477800) and EMERGE (NCT02484547), were terminated in March 2019 after a futility analysis suggested a low probability of achieving statistical significance [139]. This decision was based on the assumption of consistent treatment effects across the trials and over time, which Biogen later argued were invalid.

Subsequent analyses revealed discordant results between the trials. EMERGE met its primary endpoint, showing a 22% reduction in clinical decline in the high-dose group, while ENGAGE failed to demonstrate any clinical benefit. This inconsistency raises significant concerns about the reliability and replicability of the findings. Instead of addressing these uncertainties through a new phase III trial, Biogen pursued regulatory approval using post hoc analyses. These analyses deviated from the prespecified statistical analysis plan, failing to account for multiplicity and Type I error control. Only the high-dose group in EMERGE achieved nominal significance on the primary endpoint (P=0.012), while other doses and outcomes did not meet statistical thresholds. Despite this, Biogen characterized the results as “clinically meaningful,” even though the observed effect size on the CDR-SB scale (-0.39) was modest and inconsistent with ENGAGE.

Biogen’s reliance on amyloid plaque reduction as a surrogate endpoint further complicates the case for aducanumab. Accelerated approval was granted based on evidence of significant plaque reduction, deemed “reasonably likely” to predict clinical benefit. However, the link between amyloid clearance and cognitive improvement remains tenuous. For example, high-dose treatment in ENGAGE showed substantial plaque reduction but cognitive decline, undermining the validity of amyloid reduction as a surrogate marker.

The FDA’s decision to approve aducanumab has faced substantial criticism for prioritizing biomarker outcomes over demonstrated clinical benefits. This approach risks setting a troubling precedent and highlights the need for rigorous trial designs, adherence to prespecified analyses, and greater transparency in regulatory processes to ensure patient safety and therapeutic efficacy [140] [Schneider, 2024].

Schneider, L.S. Aducanumab Trials EMERGE But Don’t ENGAGE. J Prev Alzheimers Dis 9, 193–196 (2022). https://doi.org/10.14283/jpad.2022.37

Reviewer 1

Comment #3

While the article briefly touches on the role of peripheral immune cells (e.g., monocytes, neutrophils), it does not sufficiently explore their interactions with the central nervous system (CNS). Recent studies suggest that systemic inflammation and gut microbiota can influence AD progression through the microbiota-gut-brain axis. The authors should expand discussion on microbiota-gut-brain axis, which could provide a more comprehensive view of immune modulation.

Neuroinflammation and immune-related mechanisms are crucial in the pathophysiology and progression of AD (García-Culebras et al., 2024). The interaction between peripheral immune cells and the CNS occurs through three primary pathways: (1) the blood-brain barrier (BBB), which connects the brain to the circulation (Sweeney et al., 2018); (2) the choroid plexus (CP), which forms an interface between the blood and cerebrospinal fluid (CSF) (Dani et al., 2021); and (3) the meninges, an immune-blood-brain interface that allows immune cells to bypass the BBB and enter directly into the brain via specialized skull bone marrow channels (Herisson et al., 2018).

Myeloid cells, which are part of the innate immune system, play a key role in neuroinflammation and neurodegeneration (Doty et al., 2015). These cells include peripheral immune cells such as neutrophils and monocytes, which can infiltrate the brain and exacerbate peripheral inflammation by releasing pro-inflammatory cytokines like interleukin-1β (IL-1β) and interleukin-6 (IL-6). Lymphoid cells, such as T and B cells, are involved in adaptive immunity. While their role in AD is still not fully understood, recent studies indicate that they can infiltrate the brain (Mehdi Jorfi et al., 2023).

The integrity of the BBB is compromised by the accumulation of amyloid-β (Aβ) and tau in the AD brain and blood vessels, resulting in the release of pro-inflammatory mediators and the infiltration of myeloid cells into the brain (Sweeney et al., 2018). Furthermore, Pietronigro et al. demonstrated that neutrophil-specific protease cathepsin G accumulates in the brain and blood vessels of AD patients (Pietronigro et al., 2017), while Zenaro et al. confirmed the infiltration of neutrophils into the brains of AD mice and observed that neutrophil extracellular traps (NETs) promoted amyloid plaque formation and tau tangles, leading to worsened cognitive decline (Zenaro et al., 2015). El Khoury et al. highlighted that blood monocytes enter the brain through the vasculature, migrating toward and associating with amyloid plaques.

The role of CD4+ T cells in neurodegeneration depends on their specific subsets. Regulatory T (Treg) cells, in particular, have been linked to various neuroinflammatory and neurodegenerative diseases, including AD. Treg cells are important at the early stages of AD in regulating the clearance of β-amyloid deposits by resident microglial cells (Baruch et al., 2015; Dansokho et al., 2016). Additionally, the clonal expansion of CD8+ T cells in the brains and CSF of AD patients suggests that CD8+ T cells may impact neurodegeneration and/or cognitive impairment (Gate et al., 2020). B cells, similar to T cells, may have either protective or detrimental effects in AD, depending on multiple factors (Yang et al., 2019). Incorporating a discussion of the microbiota-gut-brain axis, as suggested, would enhance our understanding of how systemic inflammation and peripheral immune cells influence neuroinflammation and AD progression.

García-Culebras A, Cuartero MI, Peña-Martínez C, Moraga A, Vázquez-Reyes S, de Castro-Millán FJ, Cortes-Canteli M, Lizasoain I, Moro MA. Myeloid cells in vascular dementia and Alzheimer's disease: Possible therapeutic targets? Br J Pharmacol. 2024 Mar;181(6):777-798. doi: 10.1111/bph.16159. Epub 2023 Jul 7.

Sweeney MD, Sagare AP, Zlokovic BV. Blood–brain barrier breakdown in Alzheimer disease and other neurodegenerative disorders. Nat Rev Neurol. 2018;14:133–150. doi: 10.1038/nrneurol.2017.188.

Dani N, Herbst RH, McCabe C, Green GS, Kaiser K, Head JP, et al. A cellular and spatial map of the choroid plexus across brain ventricles and ages. Cell. 2021;184:3056–3074.e21. doi: 10.1016/j.cell.2021.04.003.

Herisson F, Frodermann V, Courties G, Rohde D, Sun Y, Vandoorne K, et al. Direct vascular channels connect skull bone marrow and the brain surface enabling myeloid cell migration. Nat Neurosci. 2018;21:1209–1217.

Doty KR, Guillot-Sestier MV, Town T. The role of the immune system in neurodegenerative disorders: Adaptive or maladaptive? Brain Res. 2015;1617:155–173.

Jorfi M, Maaser-Hecker A, Tanzi RE. The neuroimmune axis of Alzheimer's disease. Genome Med. 2023 Jan 26;15(1):6.

Pietronigro EC, Della Bianca V, Zenaro E, Constantin G. NETosis in Alzheimer's disease. Front Immunol. 2017;8:211.

Zenaro E, Pietronigro E, Della Bianca V, Piacentino G, Marongiu L, Budui S, et al. Neutrophils promote Alzheimer's disease-like pathology and cognitive decline via LFA-1 integrin. Nat Med. 2015;21(8):880–886.

El Khoury J, Toft M, Hickman SE, Means TK, Terada K, Geula C, Luster AD. Ccr2 deficiency impairs microglial accumulation and accelerates progression of Alzheimer-like disease. Nat Med. 2007;13(4):432–438.

Baruch K, Rosenzweig N, Kertser A, Deczkowska A, Sharif AM, Spinrad A, et al. Breaking immune tolerance by targeting Foxp3+ regulatory T cells mitigates Alzheimer's disease pathology. Nat Commun. 2015;6:7967.

Dansokho C, Ahmed DA, Aid S, Toly-Ndour C, Chaigneau T, Calle V, et al. Regulatory T cells delay disease progression in Alzheimer-like pathology. Brain. 2016;139:1237–1251.

Gate D, Saligrama N, Leventhal O, Yang AC, Unger MS, Middeldorp J, et al. Clonally expanded CD8 T cells patrol the cerebrospinal fluid in Alzheimer’s disease. Nature. 2020;577:399–404.

Yang C, Hou X, Feng Q, Li Y, Wang X, Qin L, Yang P. Lupus serum IgG induces microglia activation through Fc fragment dependent way and modulated by B-cell activating factor. J Transl Med. 2019;17:426.

Reviewer 1

Comment #4

4, The article mentions APOE ε4 as a risk factor but does not extensively discuss the implications of genetic and biomarker heterogeneity in tailoring immunotherapy. A deeper exploration of personalized approaches, such as stratifying patients based on genetic markers or cerebrospinal fluid (CSF) biomarkers, could enhance the relevance of immunotherapies.

We thank the reviewer for highlighting the important aspect of genetic and biomarker heterogeneity in Alzheimer’s disease (AD) and its implications for tailoring immunotherapies. As mentioned in the article, APOE ε4 is a well-established genetic risk factor for AD, and carriers of this allele are at higher risk of developing the disease, with more rapid progression. We agree that a deeper exploration of personalized approaches, such as stratifying patients based on genetic markers like APOE ε4 or cerebrospinal fluid (CSF) biomarkers, is essential to optimize the relevance and effectiveness of immunotherapies. As the reviewer pointed out, while APOE ε4 plays a critical role in amyloid-beta (Aβ) deposition, not all individuals with this allele develop AD, highlighting the complex interaction between genetic predisposition, environmental factors, and lifestyle. Furthermore, other genetic mutations, such as those in APP, PSEN1, and PSEN2, contribute to disease progression and could further inform the development of personalized treatment strategies. In addition to genetic markers, immune profiles, including pro-inflammatory cytokines and regulatory factors, as well as CSF biomarkers such as Aβ42, t-tau, and p-tau, provide valuable insights into the disease and enable more precise patient stratification. We also acknowledge the value of incorporating both genetic and immune markers in clinical decision-making, as these can help identify patients who are most likely to benefit from immunotherapies targeting Aβ deposition and tau aggregation. Regular monitoring of genetic and immune biomarkers will be crucial to adapt treatment plans over time and improve therapeutic outcomes, particularly in patients with mutations in drug-metabolizing genes.

We appreciate the reviewer’s thoughtful suggestions and hope this discussion helps further emphasize the importance of personalized approaches in enhancing the clinical relevance and success of immunotherapies for AD.

Morevoer, Genetic variations, such as the APOE ε4 allele, play a key role in the onset and progression of AD, with carriers showing faster disease progression. The APOE gene exists in three common forms: ε2, ε3, and ε4. While APOE ε3 is neutral, APOE ε4 increases the risk of AD, with those carrying two copies (homozygous) facing an even higher risk and earlier onset of the disease. APOE ε4 is linked to impaired amyloid-beta (Aβ) clearance in the brain, contributing to amyloid plaque buildup, a hallmark of AD. However, not everyone with APOE ε4 develops AD, highlighting the complex interaction of genetics, environment, and lifestyle in disease progression. Beyond APOE ε4, mutations in APP, PSEN1, and PSEN2 also contribute to AD. Identifying these mutations helps categorize patients into genetic subtypes, essential for personalized treatment strategies, particularly immunotherapies [Raulin et al, 2022] [Xiao et al, 2021] [Huda et al, 2023].

Immune responses, including pro-inflammatory markers like IL-1β and TNF-α, as well as regulatory factors such as IL-10, further stratify patients into immune subtypes, informing targeted therapies. Combining genetic markers and immune profiles, including cerebrospinal fluid (CSF) biomarkers, enables tailored immunotherapies, integrating treatments targeting amyloid-beta (Aβ) deposition and tau aggregation [Babić Leko et al, 2020].

CSF biomarkers such as CSF Aβ42, CSF t-tau, and CSF p-tau aid early AD diagnosis, improving diagnostic accuracy and differentiating AD from other dementias [117, 118]. While amyloid-PET is more precise, CSF biomarkers are a more accessible and cost-effective alternative. Ongoing validation of new biomarkers is critical for advancing AD diagnosis and treatment [Dubois et al, 2013][Dubois et al, 2014]. Personalized treatments should adapt to changes in genetic and immune biomarkers over time. Regular monitoring allows for adjustments in treatment plans and drug dosages, optimizing therapy and minimizing toxicity, especially in patients with mutations in drug-metabolizing genes. Considering both genetic and immune factors can refine immunotherapy approaches, enhancing the relevance and effectiveness of AD treatments tailored to individual patients.

Raulin A-C, Doss SV, Trottier ZA, Ikezu TC, Bu G, Liu C-C. ApoE in Alzheimer’s disease: pathophysiology and therapeutic strategies. Mol Neurodegener. 2022 Nov 8;17:72. doi: 10.1186/s13024-022-00574-4.

Xiao X, Liu H, Liu X, Zhang W, Zhang S, Jiao B. APP, PSEN1, and PSEN2 variants in Alzheimer’s disease: systematic re-evaluation according to ACMG guidelines. Front Aging Neurosci. 2021 Jun 18;13:695808. doi: 10.3389/fnagi.2021.695808.

Huda, T. I.; Diaz, M. J.; Gozlan, E. C.; Chobrutskiy, A.; Chobrutskiy, B. I.; Blanck, G., Immunogenomics Parameters for Patient Stratification in Alzheimer's Disease. J Alzheimers Dis 2022, 88, (2), 619-629.

Babić Leko M, Nikolac Perković M, Klepac N, Švob Štrac D, Borovečki F, Pivac N, Hof PR, Šimić G. IL-1β, IL-6, IL-10, and TNFα single nucleotide polymorphisms in humans influence the susceptibility to Alzheimer's disease pathology. J Alzheimers Dis. 2020;75(3):1029-1047.

Dubois B., Gauthier S., Cummings J. The utility of the new research diagnostic criteria for Alzheimer’s disease. Int. Psychogeriatr. 2013;25:175–177.

Dubois B., Feldman H.H., Jacova C., Hampel H., Molinuevo J.L., Blennow K., DeKosky S.T., Gauthier S., Selkoe D., Bateman R., et al. Advancing research diagnostic criteria for Alzheimer’s disease: The IWG-2 criteria. Lancet Neurol. 2014;13:614–629.

Reviewer 1

Comment #5

  1. The section on gut microbiota's role in AD is intriguing but speculative. While emerging data suggest that gut dysbiosis may contribute to neuroinflammation, the article lacks sufficient clinical evidence to support these claims robustly. The authors should include more references to longitudinal human studies or meta-analyses would strengthen this section.

Thank you for your valuable feedback. After careful consideration, we have decided to remove the section discussing the gut microbiota's role in AD. Other reviewers raised similar concerns regarding the speculative nature of this content. We believe that omitting this part will strengthen the overall manuscript and ensure a more focused discussion on well-established findings. We appreciate your understanding and input on this matter.

Reviewer 1

Comment #6

  1. 6. The discussion on tau-targeted therapies, such as semorinemab and gosuranemab, mentions trial failures without dissecting the reasons behind them. Addressing whether these failures stem from the antibody design, patient selection, or trial endpoints would provide valuable insights into the challenges of targeting tau.

We appreciate the reviewer’s valuable feedback and the suggestion to further explore the reasons behind the failures of tau-targeted therapies, such as semorinemab and gosuranemab. As noted, the semorinemab trial did not demonstrate significant improvements in cognitive or functional outcomes, despite reductions in tau biomarkers. We agree that factors such as trial design, endpoints, and the patient population (prodromal to mild AD) likely contributed to the lack of efficacy. The timing of intervention, along with the complex role of tau in AD pathology, may also have played a critical role.

Similarly, the failure of gosuranemab can be attributed to multiple factors, including the choice of targeting the N-terminus of tau, which has been associated with other unsuccessful trials. Additionally, the timing of intervention in patients with mild cognitive impairment or mild AD, as well as the use of clinical endpoints like the CDR-SB, may not have been ideal for detecting subtle disease progression.

These challenges emphasize the need for further research into optimal treatment strategies, patient selection, and trial design to improve the success of tau-targeted therapies in AD.

The semorinemab trial failed primarily because it did not show significant improvements in cognitive or functional outcomes in patients with prodromal to mild Alzheimer’s disease, despite reductions in tau biomarkers. The trial’s design, including its endpoints and patient population, may have contributed to this lack of efficacy. The timing of intervention (early-stage disease) and the complexity of tau's role in Alzheimer's pathology also likely influenced the results.

The failure of the gosuranemab (BIIB092) trial can be attributed to several factors. Despite demonstrating target engagement by lowering N-terminal tau fragments in cerebrospinal fluid (CSF), gosuranemab did not show a significant effect on tau accumulation in the brain or on cognitive decline. One key issue may have been the choice of the N-terminus of tau as the target, as other anti-tau antibodies have also failed, suggesting that this may not be the most effective approach. Additionally, the timing of intervention in patients with mild cognitive impairment or mild AD may have limited the therapeutic impact, as tau pathology may be too advanced for intervention at this stage. Furthermore, the clinical endpoints used in the study, such as the Clinical Dementia Rating-Sum of Boxes (CDR-SB), may not have been sensitive enough to detect subtle changes in disease progression, contributing to the lack of observed efficacy. These challenges highlight the complexities of targeting tau in AD and the need for further exploration of optimal treatment strategies.

Reviewer 2 Report

Comments and Suggestions for Authors

This comprehensive review is nicely organized and timely. I do have one question-

Please speculate why gamma-secretase or beta secretase inhibitors failed in clinical trials, while monoclonal antibody against Abeta peptide worked ?  

Author Response

#Reviewer 2

Please speculate why gamma-secretase or beta secretase inhibitors failed in clinical trials, while monoclonal antibody against Aβ peptide worked?  

We thank the reviewer for their thoughtful question regarding the clinical outcomes of gamma-secretase and beta-secretase inhibitors in Alzheimer’s disease (AD) trials. As discussed, while targeting amyloid-β (Aβ) has been a central approach in AD therapies, the failure of these inhibitors highlights the complexity of the disease and the unintended consequences of broad inhibition. For instance, gamma-secretase inhibitors, such as semagacestat, led to significant adverse effects by disrupting Notch signaling, a pathway critical for various physiological functions, which contributed to cognitive decline and peripheral toxicity. Additionally, beta-secretase inhibitors like verubecestat showed limited clinical efficacy despite substantial reductions in cerebrospinal fluid Aβ and modest decreases in brain amyloid load. These results raise important questions about the role of Aβ in advanced stages of the disease, suggesting that other factors may be driving disease progression.

These challenges underscore the need for more precise targeting and consideration of off-target effects when developing AD therapies. We believe that further innovation and more carefully designed clinical trials, incorporating early intervention and individualized treatment strategies, will be key to overcoming these challenges and improving therapeutic outcomes for AD patients.

We appreciate the reviewer’s insights and hope this provides a clearer understanding of the underlying reasons for the difficulties encountered in the development of gamma-secretase and beta-secretase inhibitors.

Immunotherapy for AD targeting Aβ focuses on reducing the toxic effects of Aβ plaques in the brain, which are strongly associated with the cognitive decline seen in AD. Since Aβ is thought to play a crucial role in the disease’s progression, researchers have developed different strategies to promote its clearance or prevent its accumulation. These strategies are broadly classified into two approaches: active and passive immunotherapy. Various therapeutic agents are designed to target different forms of Aβ aggregates, such as monomers, oligomers, protofibrils, fibrils, and plaques, with the goal of preventing their formation, enhancing their removal, or neutralizing their harmful effects (Figure 3). 

Despite significant efforts, many therapeutic approaches targeting Aβ have faced challenges, highlighting the complexity of intervening in amyloid pathways. Gamma-secretase and beta-secretase inhibitors, which aim to prevent Aβ production, have largely failed in clinical trials. For example, gamma-secretase inhibitors like semagacestat disrupted Notch signaling, leading to severe side effects, including cognitive decline and peripheral toxicity, due to the enzyme's role in processing multiple essential substrates. Moreover, inconsistent pharmacokinetics and transient inhibition paradoxically increased Aβ production in some cases due to oscillatory effects. Beta-secretase 1 (BACE1) inhibitors, such as Merck's verubecestat, also demonstrated limited efficacy. Even with near-maximal reductions in cerebrospinal fluid Aβ and modest decreases in brain amyloid load, verubecestat failed to slow disease progression in mild-to-moderate AD. Trials of atabecestat in asymptomatic individuals were similarly discontinued due to safety concerns, including elevated liver enzyme levels. These failures have not only tempered expectations for amyloid-centric strategies but also raised questions about the amyloid hypothesis itself, suggesting that Aβ-independent mechanisms may drive disease progression in advanced stages. 

Monoclonal antibody therapies, on the other hand, have shown greater promise in targeting Aβ. These therapies are designed to selectively bind to toxic aggregated forms of Aβ, such as plaques and oligomers, thereby reducing their pathological burden without interfering with critical physiological pathways. For instance, aducanumab and other antibodies have demonstrated potential in slowing cognitive decline, particularly when administered early in the disease course. These therapies underscore the importance of precise targeting and timing of intervention to maximize therapeutic efficacy. 

In the following sections, different types of active and passive immunotherapies based on Aβ will be explored in greater detail, highlighting their mechanisms of action, clinical outcomes, and potential future directions. 

Hur, JY. γ-Secretase in Alzheimer’s disease. Exp Mol Med 54, 433–446 (2022).

Mullard, A. BACE failures lower AD expectations, again. Nat Rev Drug Discov 17, 385 (2018).

Reviewer 3 Report

Comments and Suggestions for Authors

The manuscript submitted by Balkhi et al sets out to review the neuroimmune response in Alzheimer’s disease and the immunotherapeutic strategies aimed at modulating this response. The manuscript extensively covers the processes underlying neuroimmune responses and various therapeutics, providing a useful resource on treatments. The sections on neutrophils and NK cells, immunotherapies, and complementary approaches were particularly well written. However, the review fails to present a cohesive narrative due to extraneous details, lack of transitions, and occasional over-simplifications of biological processes and controversies. The most significant weakness of the manuscript is that the gap in the knowledge of the field is not clearly identified. A narrowing of focus and an emphasis on connecting individual facts would significantly improve readability.

General Comments:

The review is comprehensive, but the significance is not clear. The treatment section would be of relevance to the field but currently reads simply as a list of studies. A gap in knowledge is not clearly identified and there is little insight/perspective from the authors. What is there feels superficial and redundant throughout the manuscript.

No similar review has been recently published.

The cited references are appropriate; however, it would be better if in the text the references were added in the appropriate sentences/phrases, rather than grouped together at the end of the paraphs. Additionally, it should be noted not all references are formatted the same. Please use consistent formatting in the reference section. The references are timely/current (45% are more than 5 years old) and are primary research. There is not an excessive number of self-citations.

Statements and conclusions are coherent and supported by the citations. However, while technically correct, there are multiple instances where the statements are over-simplified, especially in the beginning of the review, detracting from the readability. Though a factual review, the statements/conclusions are typically of the “more needs to be done” nature, rather than making any novel or insightful conclusions about the studies referenced or the field in general.   

The figures are helpful with some minor notes. Figure 1 should either be expanded or renamed to reflect that it only addresses tau and is seemingly missing an in-text reference. Figure 2 has a typo in the description (“in contrast” alone at the end) and has a series of typos where references to the right and left panels are switched. Figure 3 doesn’t add much given the lack of details on targeting different stages.

Specific Comments:

Lines 45-50: Please provide an explanation on how amyloid beta and tau lead to pathology and clinical symptoms.

Lines 65-70, 154-163: The description of microglia oversimplifies their phenotypes, especially their metabolism, by focusing on M1 versus M2. Please revise M1/M2 to homeostatic, acute inflammatory, and DAM/MGnD microglia. When addressing metabolic differences, please include differences in lipid metabolism.

Line 116: The abbreviation “TMAO” is presented without being first defined/spelled out. Please ensure all abbreviations are defined/spelled out the first time they appear.

Lines 270-279: The impact of peripheral amyloid beta on pathology is not well explained, and as such the oversimplification could be interpreted as peripheral amyloid beta being solely responsible for activation of peripheral immune cells. Please incorporate the major effects of the CNS resident immune cells on peripheral immune cells and peripheral amyloid beta.

Lines 290-297: The section on monocytes doesn’t add much without any conclusions drawn from the authors. This section would benefit from focusing on either recruitment to the brain and surrounding interfaces, or inflammatory processes. Regarding monocyte infiltration into the brain parenchyma, please address that there is active disagreement in the field about this and include references that show monocytes do not infiltrate the brain. Some examples include Wang et al 2016, Reed 2020 (JEM), and Rivest 2013 that shows CCR2 -/- doesn’t do much

Lines 373-392: The section on complement is off-putting as there isn’t a clear transition from innate immune cells to this. Please either move this section somewhere else or remove it to help narrow the focus. Focusing on the different types of peripheral immune cells and transitioning directly from innate to adaptive cells would improve readability

Line 399: The meninges are brought up in the adaptive immune section but not mentioned in the innate immune section. Adding meninges trafficking specifically to the monocyte section would be especially helpful.

Lines 409-410: T cells are able to cross an intact BBB. Please make it clear that it is not purely breakdown of the BBB that allows T cells to infiltrate the brain parenchyma and its surrounding interfaces.

Lines 463-471: Conflicting B cell results are not really expanded on. Please elaborate on the belief in the field that B cell results seem to be influenced by experimental models.

Lines 473-503: Please expand the innate/adaptive interplay section beyond T cells and microglia and incorporate the other cells that were just discussed. Some points are oversimplified while the details mentioned do not have their significance explained. Expanding the first three paragraphs (lines 473-489) and removing the last three paragraphs (lines 490-503) would improve this section.

Lines 532-932: Different types of AD (late, early, sporadic) are mentioned throughout the immunotherapies section, most notably in the context of clinical trials. Please explain the differences between these types of AD in the introduction section.

Lines 532-932: Occasionally, results from clinical trials are just stated without interpretation, which leaves some of the significance unclear. Please ensure that all immunotherapies mentioned have a sentence or two summarizing the major findings from the clinical studies.

Line 1129: Please revise “gender differences” to “sex differences” as gender is a social construct that does not clearly exist in mice.

Lines 1188-1229: While well written, the conclusion section focuses on the roadblocks that were previously described in challenges. Please end with a summary of the overall narrative instead of that negative note to improve the significance of the manuscript.

Line 1218: Please revise the sentence on reviewed immunotherapies to reflect that they seemingly halt but do not reverse AD as they are mentioned as doing both here.

Author Response

Reviewer 3

Figure 1 should either be expanded or renamed to reflect that it only addresses tau and is seemingly missing an in-text reference. Figure 2 has a typo in the description (“in contrast” alone at the end) and has a series of typos where references to the right and left panels are switched. Figure 3 doesn’t add much given the lack of details on targeting different stages.

Thank you for your detailed feedback. We have made the following revisions to address the concerns regarding the figures:

Figure 1: While the figure content remains unchanged, the legend has been revised to better reflect its focus on tau pathology. Additionally, the in-text reference to Figure 1 has been clearly highlighted within the manuscript to ensure its location is evident.

Figure 2: The description of Figure 2 has been corrected to resolve the incomplete phrase “in contrast.” The references to the right and left panels have also been reviewed and corrected to align accurately with the figure content.

Figure 3: The original Figure 3 has been replaced with a new version that provides additional details on therapeutic strategies targeting different stages of Aβ aggregation. The revised figure also incorporates key pathological features, such as mitochondrial dysfunction, disintegrating microtubules, and neurofibrillary tangles, offering a more comprehensive depiction of AD pathogenesis.

Lines 45-50: Please provide an explanation on how amyloid beta and tau lead to pathology and clinical symptoms.

Pathologically, AD is defined by the accumulation of extracellular amyloid β-peptide (Aβ) and intracellular neurofibrillary tangles (NFTs) composed of hyperphosphorylated tau protein. Aβ and tau act synergistically to drive the progression of Alzheimer's disease, with Aβ often considered the "trigger" and tau the "bullet" in this process. Soluble Aβ species initiate tau pathology, while tau amplifies the neurotoxic effects of Aβ, leading to synaptic loss and cognitive decline. This interplay exacerbates neuronal damage, oxidative stress, synaptic dysfunction, and neuroinflammation, ultimately resulting in the loss of synapses and neurons [4-6] (Bloom et al, 2014).

Bloom GS. Amyloid-β and tau: the trigger and bullet in Alzheimer disease pathogenesis. JAMA Neurol. 2014;71:505–508.

Microglial activation plays a significant role in mediating these pathological effects. Oligomeric Aβ promotes microglial activation, which contributes to early synaptic loss. Moreover, microglia may facilitate the spread of tau pathology, rendering synapses more vulnerable to tau-induced impairments. This neuroinflammatory cascade leads to neuronal cell death, plasma leakage, and brain atrophy. These pathological changes disrupt communication between brain cells, impairing cognitive functions such as memory and reasoning (Bloom et al, 2014)(Alawieyah et al, 2018).

Bloom GS. Amyloid-β and tau: the trigger and bullet in Alzheimer disease pathogenesis. JAMA Neurol. 2014;71:505–508.

Alawieyah Syed Mortadza S, Sim JA, Neubrand VE, Jiang LH. A critical role of TRPM2 channel in Aβ42-induced microglial activation and generation of tumor necrosis factor-α. Glia. 2018;66:562–575.

A potential link between amyloid and tau pathology may also involve the innate immune system, with both proteins triggering immune responses that further propagate damage. Together, these processes create a cascade of neuronal dysfunction and degeneration that underpins the clinical symptoms of AD (Roda et al, 2022).

Roda AR, Serra-Mir G, Montoliu-Gaya L, Tiessler L, Villegas S. Amyloid-beta peptide and tau protein crosstalk in Alzheimer's disease. Neural Regen Res. 2022;17(8):1666-1674.

Lines 65-70, 154-163: The description of microglia oversimplifies their phenotypes, especially their metabolism, by focusing on M1 versus M2. Please revise M1/M2 to homeostatic, acute inflammatory, and DAM/MGnD microglia. When addressing metabolic differences, please include differences in lipid metabolism.

Thank you for your thoughtful feedback. We appreciate your suggestion to revise the discussion of microglial phenotypes and metabolism to provide a more nuanced and accurate representation. In response, we have updated the manuscript as follows:

Microglial Phenotypes:

We have revised the text to replace the oversimplified M1/M2 paradigm with the more current classification of homeostatic, acute inflammatory, and disease-associated microglia (DAM)/microglia in neurodegenerative diseases (MGnD). This updated framework better reflects the complexity of microglial states and their dynamic roles in AD progression.

Metabolic Differences:

The description of microglial metabolism has been expanded to include lipid metabolism, which is a critical aspect of DAM/MGnD function. The revised text highlights the shift toward increased lipid biosynthesis and remodeling in DAM/MGnD microglia, linking this metabolic reprogramming to their pro-inflammatory and neurodegenerative roles. Additionally, we have clarified the metabolic reliance of homeostatic microglia on oxidative phosphorylation and the TCA cycle, as well as the glycolytic preference of acute inflammatory microglia, drawing parallels to the "Warburg effect."

We believe these changes provide a more comprehensive and accurate depiction of microglial biology in AD and address the concern raised. Thank you for pointing out this important area for improvement.

Line 116: The abbreviation “TMAO” is presented without being first defined/spelled out. Please ensure all abbreviations are defined/spelled out the first time they appear.

This belonged to the gut microbiota section that has been deleted.

Lines 270-279: the major effects of the CNS resident immune cells on peripheral immune cells and peripheral amyloid beta

The section is added to the manuscript.

Lines 290-297: The section on monocytes doesn’t add much without any conclusions drawn from the authors. This section would benefit from focusing on either recruitment to the brain and surrounding interfaces, or inflammatory processes. Regarding monocyte infiltration into the brain parenchyma, please address that there is active disagreement in the field about this and include references that show monocytes do not infiltrate the brain. Some examples include Wang et al 2016, Reed 2020 (JEM), and Rivest 2013 that shows CCR2 -/- doesn’t do much

We thank the reviewer for their thoughtful comments. We have revised the section to draw more focused conclusions regarding the recruitment of monocytes and their role in inflammatory processes in Alzheimer’s disease (AD). Furthermore, we have addressed the active debate on monocyte infiltration into the brain parenchyma and provided evidence from Reed (2020), Wang (2016), and Rivest (2013) to clarify this issue.

Lines 373-392: The section on complement is off-putting as there isn’t a clear transition from innate immune cells to this. Please either move this section somewhere else or remove it to help narrow the focus. Focusing on the different types of peripheral immune cells and transitioning directly from innate to adaptive cells would improve readability

The complement section is removed.

Line 399: The meninges are brought up in the adaptive immune section but not mentioned in the innate immune section. Adding meninges trafficking specifically to the monocyte section would be especially helpful.

Thank you for your valuable suggestion. We have now included the information regarding meninges trafficking specifically in the monocyte section. The revised text now emphasizes that, under homeostatic conditions, monocytes originating from the skull bone marrow infiltrate the meninges via specialized vascular channels, where they differentiate into macrophages and dendritic cells (DCs). This addition highlights the unique role of the meninges in immune cell trafficking and differentiation, both in steady-state and disease conditions, including neuroinflammation. The updated section now reflects this crucial aspect of monocyte migration and its impact on the immune landscape, particularly in the context of neuroinflammatory diseases such as AD. We believe this addition strengthens the connection between innate immune mechanisms and the role of the meninges in immune surveillance.

Monocytes can also infiltrate the meninges under both steady-state and inflammatory conditions, with skull bone marrow serving as a key reservoir for monocyte precursors. These cells, upon entering the dura via specialized vascular channels, can differentiate into antigen-presenting cells such as dendritic cells or into macrophages. This meningeal trafficking provides additional immune surveillance and modulation, particularly in disease states associated with neuroinflammation. Enhanced monocyte recruitment to the meninges has been observed in models of neuroinflammation, highlighting the potential for meningeal immune responses to influence AD pathology.

Lines 409-410: T cells are able to cross an intact BBB. Please make it clear that it is not purely breakdown of the BBB that allows T cells to infiltrate the brain parenchyma and its surrounding interfaces.

The part is changed based on the comment.

Lines 463-471: Conflicting B cell results are not really expanded on. Please elaborate on the belief in the field that B cell results seem to be influenced by experimental models.

We thank the reviewer for highlighting this important point. Conflicting findings regarding B cell involvement in AD likely stem from differences in experimental models. While tau-specific antibodies are detected in AD patients, studies in tauopathy mice show mixed results, with some indicating minimal B cell influence on tau pathology, while others report that genetic B cell depletion exacerbates spatial memory deficits. These discrepancies may arise from variations in genetic backgrounds, depletion methods, or the limited ability of murine models to replicate human immune responses. Further studies using more representative models are needed to better understand the role of B cells in AD.

However, conflicting results have emerged regarding the impact of B cell depletion on tau pathology in AD. While tau-specific antibodies are detected in the brain and serum of AD individuals, studies in tauopathy mice suggest that B cells do not influence tau pathology. Nevertheless, genetic B cell depletion in tau transgenic mice has been shown to moderately exacerbate spatial memory deficits [120].

The precise role of B cells in AD etiology and progression remains unclear, with conflicting findings necessitating further investigation. Understanding the nuanced interactions between B cells, other immune cells, and neuronal components in the context of AD pathology is crucial for developing targeted therapeutic strategies.

Lines 473-503: Please expand the innate/adaptive interplay section beyond T cells and microglia and incorporate the other cells that were just discussed. Some points are oversimplified while the details mentioned do not have their significance explained. Expanding the first three paragraphs (lines 473-489) and removing the last three paragraphs (lines 490-503) would improve this section.

We couldn’t find references that talks beyond the role of T cells and microglia in this interaction.

Lines 532-932: Different types of AD (late, early, sporadic) are mentioned throughout the immunotherapies section, most notably in the context of clinical trials. Please explain the differences between these types of AD in the introduction section.

We thank the reviewer for their insightful comment. To address the request, the differences between the main types of AD have been included in the introduction to provide a clearer context for the discussion in the immunotherapies section.

The classification of AD includes Early-Onset Alzheimer’s Disease (EOAD), Late-Onset Alzheimer’s Disease (LOAD), and sporadic Alzheimer’s Disease. These categories are defined based on the age at which the disease manifests and the associated genetic factors. EOAD is diagnosed before the age of 65 and accounts for less than 5% of all Alzheimer’s cases. It is often associated with genetic mutations in the APP, PSEN1, and PSEN2 genes, which lead to the accumulation of amyloid-beta plaques and result in rapid cognitive decline. Symptoms commonly include aphasia and executive dysfunction. In contrast, LOAD is the most prevalent form of AD, typically developing after the age of 65. LOAD is characterized by a slower progression, with memory deficits being the primary symptom. It is strongly linked to the APOE ε4 allele as a significant genetic risk factor.  Sporadic AD, which is generally considered synonymous with LOAD, arises from a combination of aging, lifestyle factors, and comorbidities such as cardiovascular disease. Unlike EOAD, it does not exhibit a clear familial inheritance pattern but shares similar pathological features, including amyloid-beta plaques, tau tangles, and neuroinflammation. Together, these variations highlight the complex interplay between genetic factors, environmental influences, and neurodegenerative processes central to the pathophysiology of AD.

Barber RC. The genetics of Alzheimer's disease. Scientifica (Cairo). 2012;2012:246210. doi: 10.6064/2012/246210.

Andrade-Guerrero J, Santiago-Balmaseda A, Jeronimo-Aguilar P, et al. Alzheimer's Disease: An Updated Overview of Its Genetics. Int J Mol Sci. 2023;24(4):3754.

Line 1129: Please revise “gender differences” to “sex differences” as gender is a social construct that does not clearly exist in mice.

It is changed.

Lines 532-932: Occasionally, results from clinical trials are just stated without interpretation, which leaves some of the significance unclear. Please ensure that all immunotherapies mentioned have a sentence or two summarizing the major findings from the clinical studies.

Thank you for your feedback. We've reviewed lines 532-932 and identified instances where the results from clinical trials were stated without interpretation. To address this, we've added brief explanations summarizing the major findings for each immunotherapy mentioned. This should provide clearer insight into the significance of the clinical studies.

Prof Lines 1188-1229: While well written, the conclusion section focuses on the roadblocks that were previously described in challenges. Please end with a summary of the overall narrative instead of that negative note to improve the significance of the manuscript.

Thank you for your insightful feedback regarding the conclusion section. We agree that ending the manuscript on a more positive and forward-looking note would improve its impact. In light of your comments, we have revised the conclusion to place greater emphasis on the overall narrative and the potential significance of ongoing advancements in immunotherapy for AD. The updated conclusion now highlights the transformative potential of combining advanced preclinical models, innovative therapeutic strategies, and personalized approaches to AD treatment. It underscores the optimism in the field by reflecting on how these efforts are paving the way toward disease-modifying therapies and the broader implications for neurodegenerative diseases. We have ensured that the revised section shifts the focus from roadblocks to progress and opportunities, as suggested. We hope this addresses your concern effectively and strengthens the overall significance of the manuscript. Thank you for your valuable input.

Line 1218: Please revise the sentence on reviewed immunotherapies to reflect that they seemingly halt but do not reverse AD as they are mentioned as doing both here.

The development of these advanced immunotherapeutic strategies offers hope for halting the progression of AD, though reversing the disease remains unachieved, and also holds broader implications for the treatment of other neurodegenerative diseases. With ongoing research, it is becoming increasingly clear that the immune system plays a central role in neurodegeneration, and as we gain more insights into these processes, the possibility of developing disease-modifying therapies becomes more realistic.

Reviewer 4 Report

Comments and Suggestions for Authors

In this paper, these authors review involvement of neuroinflammatory processes in the pathology of Alzheimer’s disease (AD). The paper focuses on recent advances in immunotherapy strategies aimed at modulating immune responses in AD with an emphasis on microglial behaviour, amyloid clearance, and tau tangle pathology. The aim is to explore the potential of immunotherapeutic approache to alter disease progression and improve patient outcomes. The review starts by introducing the amyloid cascade hypothesis followed by microglial phenotypes and their initial role in clearing amyloid plaques and later role in releasing toxic cytokines and inflammatory factors. A possible role of changes in the gut microbiome in AD that could drive pathology are presented and how Aβ oligomers and tau tangles can activate microglial while, conversely,  microglia can generate Aβ oligomers and tau tangles. The role of the peripheral immune system in influencing brain inflammatory pathways is then debated in some detail. The review then moves on to current and potentially novel active and passive immunotherapies for AD. To date, active and passive immunotherapy  targeting extracellular Aβ and tau aggegates has failed to improve cognition in AD despite three agents receiving temporary FDA licenses.  Novel approaches targeting microglial receptors, altering the gut microbiome, stem cell therapy, and the use of phytotherapy are debated. Development of nanoparticles to transport immunotherapies across the blood-brain barrier is ialso ntroduced. It is concluded that current animal AD models are not fit for purpose and that there is a pressing need for detailed studies that investigate optimal dosing regimens, timing of treatment administration, and the characteristics of antibodies  produced which is crucial for maximising therapeutic effectiveness and ensuring meaningful clinical outcomes.

This is a detailed review which provides several helpful figures. It is overlong and some sections, such as the possible roles of stem cell implants and phytotherapies  to potentially treat AD do not directly concern immunomodulation and could be dropped. Similarly, the sections on the role and the alteration of the gut microbiome in causing and treating AD is off message and highly speculative and could also go. An issue for me was that there is an assumption in this review that targeting extracellular Aβ aggregates  and the associated immunochanges is the correct way forward but, if that were true, anti-Aβ immunoglobins would have improved cognition. It has been proposed that raised intraneuronal Aβ and its aggegates are probably the true villain as this triggers the intracellular amyloid cascade. In summary, this review is detailed, highly instructive and educational but needs to be more focussed.  

Author Response

Reviewer 4

An issue for me was that there is an assumption in this review that targeting extracellular Aβ aggregates and the associated immunochanges is the correct way forward but, if that were true, anti-Aβ immunoglobins would have improved cognition. It has been proposed that raised intraneuronal Aβ and its aggregates are probably the true villain as this triggers the intracellular amyloid cascade.

Thank you for your insightful comment. We agree that intraneuronal Aβ plays a critical role in Alzheimer’s disease pathogenesis, preceding extracellular plaque deposition. Various studies show that Aβ42 accumulates within organelles such as the endoplasmic reticulum and endosomal-lysosomal systems, leading to synaptic degradation, impaired axonal transport, and structural damage. Intraneuronal Aβ also influences tau pathology and may act as a precursor to neurofibrillary tangles. Importantly, the interplay between intracellular and extracellular Aβ suggests a bidirectional process, where intraneuronal Aβ seeds extracellular plaques and extracellular Aβ exacerbates intracellular accumulation. These findings highlight the need for therapeutic strategies targeting both intra- and extracellular Aβ.

The Role of Intraneuronal Amyloid-Beta in AD

Recent studies suggest that intraneuronal accumulation of Aβ also plays a critical role in the pathogenesis of AD. While extracellular Aβ plaques are long-recognized pathological hallmarks, increasing evidence highlights that Aβ aggregates within neurons before extracellular plaque formation, contributing to early synaptic dysfunction and cognitive decline.

Aβ peptides, particularly Aβ42, accumulate intracellularly in organelles such as the endoplasmic reticulum, Golgi apparatus, and endosomal-lysosomal systems (Hartmann et al, 1997) (Xu et al, 1997)(Pacheco-Quinto et al, 2013). This intraneuronal Aβ is associated with disrupted neuronal function, including synaptic degradation, impaired axonal transport, and loss of microtubule-associated proteins like MAP2. Notably, Aβ oligomers, the soluble precursors of plaques, are particularly neurotoxic, causing structural alterations in synapses and neurites (Iqbal et al, 2009).

Immunoelectron microscopy studies have demonstrated that intraneuronal Aβ localizes within multivesicular bodies and other vesicles, suggesting a role in the dysregulation of cellular trafficking. Additionally, intracellular Aβ appears to influence tau pathology, potentially acting as a precursor to neurofibrillary tangles (Almeida et al, 2006)(Gruenberg et al, 2004).

The interplay between intracellular and extracellular Aβ is significant. Intraneuronal Aβ may seed extracellular plaque formation through neuritic degeneration, while extracellular Aβ can be internalized, exacerbating intracellular accumulation. These findings challenge the traditional focus on extracellular plaques and suggest that targeting intraneuronal Aβ might provide novel therapeutic opportunities.

Hartmann T, Bieger SC, Bruhl B et al. Distinct sites of intracellular production for Alzheimer's disease Ab40/42 amyloid peptides. Nat Med 1997; 3: 1016–20.

Xu H, Sweeney D, Wang R et al. Generation of Alzheimer b-amyloid protein in the trans-Golgi network in the apparent absence of vesicle formation. Proc Natl Acad Sci USA 1997;94: 3748–52.

Pacheco-Quinto J, Eckman EA. Endothelin-converting enzymes degrade intracellular b-amyloid produced within the endosomal/lysosomal pathway and autophagosomes. J Biol Chem 2013; 288: 5606–15.

Iqbal K, Liu F, Gong C-X et al. Mechanisms of tau-induced neurodegeneration. Acta Neuropathol 2009; 118: 53–69.

Almeida CG, Takahashi RH, Gouras GK. b-amyloid accumulation impairs multivesicular body sorting by inhibiting the ubiquitin-proteasome system. J Neurosci 2006; 26: 4277–88.

Gruenberg J, Stenmark H. The biogenesis of multivesicular endosomes. Nat Rev Mol Cell Biol 2004; 5: 317–23.

Round 2

Reviewer 2 Report

Comments and Suggestions for Authors

I recommend Acceptance

Reviewer 3 Report

Comments and Suggestions for Authors

The extensive revisions are greatly appreciated, resulting in a vastly improved manuscript. The only minor suggestion is to change the section title "Interplay between innate and adaptive immunity in AD" to something like "Immune microenvironment in the brain parenchyma" as that would be more representative of what is discussed.